# The Past, Present, and Future of Plant Activators Targeting the Salicylic Acid Signaling Pathway

**DOI:** 10.3390/genes15091237

**Published:** 2024-09-23

**Authors:** Misbah Naz, Dongqin Zhang, Kangcen Liao, Xulong Chen, Nazeer Ahmed, Delu Wang, Jingjiang Zhou, Zhuo Chen

**Affiliations:** 1State Key Laboratory of Green Pesticides, Key Laboratory of Green Pesticide and Agricultural Bioengineering, Ministry of Education, Guizhou University, Guiyang 550025, China; misbahnaz.ray@yahoo.com (M.N.); gs.kcliao22@gzu.edu.cn (K.L.); xulongc817@126.com (X.C.); jjzhou@gzu.edu.cn (J.Z.); 2College of Forestry, Guizhou University, Guiyang 550025, China; dlwang@gzu.edu.cn

**Keywords:** plant activators, salicylic acid signaling pathway, systemic acquired resistance, plant disease resistance, mode of action

## Abstract

Plant activators have emerged as promising alternatives to conventional crop protection chemicals for managing crop diseases due to their unique mode of action. By priming the plant’s innate immune system, these compounds can induce disease resistance against a broad spectrum of pathogens without directly inhibiting their proliferation. Key advantages of plant activators include prolonged defense activity, lower effective dosages, and negligible risk of pathogen resistance development. Among the various defensive pathways targeted, the salicylic acid (SA) signaling cascade has been extensively explored, leading to the successful development of commercial activators of systemic acquired resistance, such as benzothiadiazole, for widespread application in crop protection. While the action sites of many SA-targeting activators have been preliminarily mapped to different steps along the pathway, a comprehensive understanding of their precise mechanisms remains elusive. This review provides a historical perspective on plant activator development and outlines diverse screening strategies employed, from whole-plant bioassays to molecular and transgenic approaches. We elaborate on the various components, biological significance, and regulatory circuits governing the SA pathway while critically examining the structural features, bioactivities, and proposed modes of action of classical activators such as benzothiadiazole derivatives, salicylic acid analogs, and other small molecules. Insights from field trials assessing the practical applicability of such activators are also discussed. Furthermore, we highlight the current status, challenges, and future prospects in the realm of SA-targeting activator development globally, with a focus on recent endeavors in China. Collectively, this comprehensive review aims to describe existing knowledge and provide a roadmap for future research toward developing more potent plant activators that enhance crop health.

## 1. Introduction

To defend against pathogen attacks, plants have evolved two primary types of defense mechanisms: PAMP-triggered immunity (PTI), which relies on pathogen-associated molecular patterns (PAMPs), and effector-triggered immunity (ETI), which involves nucleotide-binding leucine-rich repeat (NLR) proteins [1]. Systemic acquired resistance (SAR) is an inducible, whole-plant immune response in which localized exposure to a pathogen or an elicitor confers long-lasting, broad-spectrum resistance against subsequent infections, thereby helping to control plant diseases. SAR involves the activation of defense genes and the production of pathogenesis-related (PR) proteins, which contribute to the plant’s heightened resistance. Notably, SAR provides broad-spectrum resistance not only against the initial pathogen but also against various other pathogens, including viruses, bacteria, and fungi. This resistance is long-lasting and can be maintained for extended periods, offering sustained protection [2]. In recent years, advances in science and technology have led to a better understanding of plant immunity, allowing for the broader application of these principles in crop disease control [1,2]. Plant immunity is typically induced by exogenous elicitors. Based on the characteristics of induced resistance, scientists have identified and developed plant activators capable of enhancing plant immunity to inhibit pests and pathogens [3]. Currently, plant activators include sugars, glycopeptides, lipids, lipopeptides, proteins, low-molecular-weight metabolites, and low-molecular-weight synthetic compounds. This review discusses how plants perceive conserved microbial signatures through pattern recognition receptors (PRRs) and explores the role of these receptors in activating innate immune responses and their implications for plant immunity [4]. The article reviews the function and discovery of PRRs and their exploitation for enhancing broad-spectrum disease resistance. It highlights advances in understanding PRR-mediated immune responses and their applications in agriculture, while also discussing how proteomic approaches can reveal the molecular mechanisms underlying plant defense responses [3,4,5,6].

When plant activators bind to PRRs, the plant immune response system is activated. PRRs include receptor-like kinases (RLKs) and receptor-like proteins (RLPs). RLKs are located on the plasma membrane and consist of an extracellular ligand-binding domain, a transmembrane domain, and an intracellular protein kinase domain. In contrast, RLPs have only extracellular and transmembrane domains, lacking the intracellular kinase domain of RLKs. Therefore, RLPs require interaction with other intracellular co-receptor kinases to transmit downstream signals [3]. Central to systemic acquired resistance (SAR) is the salicylic acid (SA) signaling cascade, a pivotal molecular pathway that orchestrates a multifaceted defense program upon activation [1,2]. This pathway prominently features the accumulation of pathogenesis-related proteins (PRs), along with the potentiation of enzymes involved in oxidative burst and other frontline defense responses (Figure 1) [7,8]. Over the past four decades, intensive research has elucidated the intricate network architecture, key components, and regulatory mechanisms governing the SA signaling pathway [8]. For instance, critical players such as the harpin-binding protein (HrBP1), enhanced disease susceptibility (EDS1 and EDS5) proteins, the master regulator nonexpressor of pathogenesis-related proteins 1 (NPR1), WRKY transcription factors, and the PRs themselves have been identified as essential signaling hubs, amplifiers, and executors of the SA-mediated resistance response [1,2,9,10].

Capitalizing on this knowledge, researchers and agrochemical companies have actively pursued the development of synthetic “plant activators” capable of stimulating the SA signaling pathway and conferring robust, broad-spectrum disease resistance without directly targeting pathogens (Figure 1). The remarkable ability of plants to “immunize” themselves against potential threats was first reported by Ross in 1961, who observed that localized infection of tobacco leaves with tobacco mosaic virus (TMV) prevented the systemic spread of the same virus upon subsequent inoculation [2,11,12]. Subsequent pioneering work by Malamy et al. [13] and Métraux et al. [14] demonstrated that systemic acquired resistance (SAR) is mediated by a long-distance signal produced at the initial infection site, which travels through the vascular system to activate defense responses in distal, uninfected parts of the plant. A major breakthrough came with the identification of salicylic acid (SA) as a crucial signaling molecule required for the establishment of SAR [13,15]. Ward et al. [16] highlighted the pivotal role of PRs as antimicrobial proteins, the expression of which is tightly regulated by the SA signaling pathway. Further dissection of the SA signaling cascade identified key regulatory components, such as the transcriptional co-activator NPR1, which acts as a master regulator of SAR by orchestrating the expression of PR genes and other defense responses upon perception of the SA signal [17,18]. Subsequent studies revealed additional players, including the EDS1 and PAD4 lipase-like proteins, which function upstream of SA biosynthesis, and the WRKY transcription factor family, which acts downstream of NPR1 to regulate *PR* gene expression [19,20,21,22].

The elucidation of the SA-mediated systemic acquired resistance (SAR) pathway has significant implications for crop protection strategies [23]. Researchers have developed synthetic “plant activators” that can prime the SA signaling pathway, thereby conferring broad-spectrum disease resistance without directly targeting pathogens [24]. This approach offers several advantages over conventional pesticides and has led to the successful commercialization of activators such as benzothiadiazole for widespread use on crops [25]. The benefits of these activators include prolonged resistance against pathogens, lower spraying dosages, reduced environmental impact, and minimal risk of pathogen resistance development [26]. Consequently, numerous SA-targeting activators, including benzothiadiazole derivatives, have been successfully commercialized and widely adopted for global crop protection [27]. Additionally, plant activators have served as important chemical probes, aiding in the elucidation of mechanisms related to the SA signaling pathway and associated physiological phenomena [28].

In this review, we trace the historical development of plant activators and elucidate the current understanding of the SA signaling pathway’s architecture and regulatory mechanisms. Furthermore, we describe the structural features, bioactivities, proposed modes of action, and field performance of both classical and emerging SA-targeting activators (Figure 2). Finally, we highlight the current challenges and future prospects in the field and introduce recent research in China aimed at advancing the development of more potent plant activators.

## 2. Classification of Plant Activators

Plant activators can be classified based on their mechanisms of action and chemical structures, each influencing the SA signaling pathway in different ways. Scientific studies have extensively explored these classifications, providing insights into how different activators modulate plant defense mechanisms. For instance, activators that act upstream of the SA signaling pathway, such as harpin-binding protein 1 (HrBP1) and β-aminobutyric acid (BABA), stimulate early defense responses before SA production (Figure 3a), as evidenced by their roles in initiating immune activity [29,30,31]. Conversely, activators acting downstream of the SA signaling pathway, such as benzothiadiazole (BTH) and 2,6-dichloroisonicotinic acid (INA), directly engage with SA-dependent pathways, inducing resistance through direct modulation of defense gene expression [32,33,34]. Additionally, studies on activators with unclear action sites, such as isotianil, have shed light on their modes of action [35,36,37]. By classifying plant activators, researchers can better understand their functions and optimize their applications in agriculture.

### 2.1. Classification by Mechanism of Action

Plant activators can be categorized based on their mode of action, affecting different steps of the SA signaling pathway [31,32,33,34]. Upstream activators, such as HrBP1 and BABA, initiate defense responses prior to SA production and subsequently increase SA accumulation [29,30,31,38]. In contrast, downstream activators, such as BTH and INA, directly enhance SA-mediated defense responses without influencing SA accumulation. Research has shown that BTH effectively mimics SA, leading to increased expression of defense-related genes and improved resistance against pathogens [32,39]. Activators like isotianil pose challenges for researchers due to their unclear modes of action [36]. Understanding these mechanisms is crucial for developing effective control strategies against crop diseases using such activators.

#### 2.1.1. Upstream Activators

Upstream activators are compounds that affect the early steps of the SA signaling pathway, prior to the production of SA itself [40].

##### HrBP1

To date, there has been limited research published on HrBP1 and its function [29,41]. HrBP1 was first isolated from a cDNA library of *Arabidopsis thaliana*. This protein is localized to the plant cell wall and, when activated, can prime various signaling pathways involving salicylic acid (SA), ethylene (ET), and others. HrBP1 is involved in forming ion channels in the plant cell wall and plays a role in signal transduction [29]. The sequence of HrBP1 across different species shows that it is highly conserved evolutionarily [29]. Harpins are secreted by Gram-negative phytopathogenic bacteria with hypersensitive response and pathogenicity (hrp) gene clusters, which are controlled by three types of secretion channels [29,30,42]. These proteins are characterized by glycine- or cysteine-rich sequences and exhibit high thermal and proteinase stability. Harpin was first isolated from *Erwinia amylovora*, a bacterial pathogen causing fire blight in rosaceous plants. It was subsequently identified as a hypersensitive-response (HR) elicitor in *E. amylovora* and later found in *Ellatocystis chrysanthemi*, *Pseudomonas syringae*, *Ralstonia solanacearum*, and *Xanthomonas* sp., where it was named hrpNEch, hrpZ, popA, and harpin-like Hpa1, respectively [43,44]. Spraying harpin on the fruits or leaves of apple trees can enhance plant resistance and reduce disease severity from *Penicillium expansum* [45]. Spraying harpin on citrus fruit can reduce the number of lesions from citrus black spot and decrease the pycnidia count produced by the pathogen [46]. On tomato leaves, harpin enhances peroxidase (POD) activity, reduces disease severity caused by tomato blight (*Phytophthora infestans*), and inhibits the growth of *P. syringae* pv. tomato [47]. Pretreatment with harpin also reduces the number of lesions from citrus leaf diseases caused by *Xanthomonas axonopodis* pv. citrumelo and *X. axonopodis* pv. citri [48]. Additionally, spraying harpin without pathogen inoculation promotes early plant growth. For apple fruit blue mold caused by *P. expansum*, pretreatment with harpin increases cell wall density. Morphological observations show tannin accumulation in vacuoles and at appositions in the epidermal and hypodermal cells of apple fruits, which inhibits mycelial growth [45]. Harpins can cause plant cell alkalinization, generate an oxidative burst through reactive oxygen species (ROS) accumulation, and induce apoptosis in plant cells when subsequently infected by pathogens [30,49,50]. Harpin Psph can induce the expression of the *HIN1* gene, part of the tobacco *PR* gene family, as well as the *8-kD salicylic acid-responsive mitogen-activated protein kinase* (*MAPK*) gene [51]. Harpin activates the SA signaling pathway by increasing *PR-1* mRNA levels and SA concentration [52]. It can also bind to HrBP1 to form a protein–protein complex, which researchers speculate undergoes a spatial structural change to activate the SA signaling pathway [53]. HrpZpss binds to the cell wall of tobacco suspension cells or tobacco leaf cells but not to protoplasts, indicating that the action site is located on the plant cell wall [54].

##### Validamycin A (VMA) and Validoxylamine A (VAA)

These antibiotics, isolated from the bacterium *Streptomyces hygroscopicus* var. *limoneus*, are known to induce SAR in plants [55]. They function by triggering the SA signaling pathway, thereby increasing resistance against a variety of pathogens [56]. VMA ([1S-(1α, 4α, 5β, 6α)]-1, 5, 6-trideoxy-4-O-β-D-glucopyranosyl-5-(hydroxymethyl)-1-[[4, 5, 6-trihydroxy-3-(hydroxymethyl)-2-cyclohexen-1-yl]amino]-D-chiro-inositol-1L-(1, 3, 4/2, 6)-2, 3-dihydroxy-6-hydroxymethyl-4-[(1S, 4R, 5S, 6S)-4, 5, 6-trihydroxy-3-hydroxymethylcyclohex-2-enylamino]cyclohexyl-3-D-glucopyranoside) is an aminoglucoside compound (Figure 3b). The first aminoglycoside, streptomycin, was isolated from Streptomyces griseus in 1943, and neomycin was isolated from *Streptomyces fradiae*. VMA is highly effective against various bacterial infections [55]. VAA ([1S-(1α, 4α, 5β, 6α)]-1, 5, 6-trideoxy-5-(hydroxymethyl)-1-[[4, 5, 6-trihydroxy-3-(hydroxymethyl)-2-cyclohexen-1-yl]amino]-D-chiro-inositol) is an aglycone of VMA [57]. The mechanism of pathogen control by VMA and VAA involves two action modes: direct inhibition and induction of plant resistance. For example, VMA can inhibit fungal growth in vitro by decreasing trehalase activity [58]. Pretreatment with VMA and VAA reduces disease severity in tomatoes infected by *Pseudomonas solanacearum* and cabbages infected with *Xanthomonas campestris* pv. *campestris* or *Fusarium oxysporum* f. sp. *lycopersici*. These results indicate that VMA and VAA possess resistance-inducing activity [59,60]. VMA and VAA can increase the levels of effector markers in the SA signaling pathway in tomato leaves, such as PR-1, PR-2, and PR-5. Their inductive effects are similar to those of BTH. VMA and VAA also enhance the concentration of free SA in both sprayed and younger (higher) leaves of tomatoes. After treatment with these compounds, the amount of free SA in the upper leaves of sprayed plants was significantly higher than in the sprayed leaves. In terms of SA concentration in the upper leaves, the resistance-inducing ability of VMA and VAA was significantly greater than that of PBZ treatment. These results suggest that VMA and VAA act upstream of SA in the SA signaling pathway [61,62].

##### Probenazole (PBZ)

PBZ (3-allyloxy-1, 2-benzisothiazole-1, 1-dioxide) is a synthetic plant activator widely used in agriculture, particularly in Japan, to control rice blast disease caused by the filamentous fungus *Magnaporthe oryzae* (Figure 3c) [63]. Developed as a fungicide during the 1970s and 1990s (trade name: Oryzaemate), PBZ is widely used to prevent and control rice blast and rice bacterial blight in Asia [63]. It functions by stimulating the SA signaling pathway, which enhances the plant’s natural defense mechanisms and increases resistance to the rice blast pathogen. More recently, PBZ has also been shown to act as an activator, inducing increased plant resistance against TMV, *P. syringae* pv. *tomato* (*Pst*), and *Oidium* sp. [33,36,64].

PBZ can upregulate the expression of several defense-related genes in rice plants, such as those encoding POD, chitinase, and PRs. It also induces the accumulation of *PR-1*, *PR-2*, and *PR-5* mRNAs in *A. thaliana* leaves, with PR-1 increasing in a dose-dependent manner. Additionally, PBZ enhances the activity of defense-related enzymes, such as phenylalanine ammonia-lyase (PAL) and POD. However, PBZ does not induce SA accumulation or upregulate *PR-1* mRNA abundance in transgenic *A. thaliana* plants expressing *salicylate hydroxylase* (*NahG*) or the signaling mutant *npr1-1* [65,66,67]. Furthermore, PBZ does not exhibit an inductive effect on disease resistance in *NahG* transgenic tobacco. These results indicate that PBZ acts upstream of SA in the SA signaling pathway [68]. The molecule 1, 2-benzisothiazol-3 (2H)-1, 1-dioxide (BIT), also known as the first artificial sweetener saccharin, has been identified as the active metabolite of PBZ (Figure 3d) [68]. Interestingly, PBZ can induce *PBZ1* mRNA, which is equivalent to *PR-10* mRNA. Although the expression of *PBZ1* mRNA is induced by the rice blast fungus, it is not correlated with rice blast resistance [66,67].

##### (D, L)-3-Aminobutyric Acid (BABA)

BABA is a non-protein amino acid that induces disease resistance in plants by activating the SA signaling pathway. It primes the plant’s immune system, leading to a faster and stronger defense response to subsequent pathogen attacks. BABA was first found to increase the resistance of pea plants to the oomycete *Aphanomyces euteiches* [69,70]. This study revealed that treatment with BABA can lead to the accumulation of β-1,3-glucanases, chitinases, and endogenous salicylic acid in pepper plants infected by Phytophthora blight [70]. Subsequently, BABA was found to induce resistance to a variety of plant diseases caused by fungal pathogens. BABA treatment can also lead to the accumulation of PRs in the intercellular fluid extracted from *Nicotiana tabacum* L. leaves [71]. The abundance of *PR-1* mRNA was enhanced by BABA treatment, indicating that BABA acts as a systemic inducer [70]. Furthermore, BABA triggers a significant increase in the endogenous levels of SA in pepper stems, with a more pronounced increase observed in *P. capsica*-infected stems following BABA treatment. These results suggest that BABA acts upstream in the SA signaling pathway [70].

##### Imprimatins

These synthetic compounds are known to enhance plant immunity by activating the SA signaling pathway. Imprimatins increase SA levels and promote the expression of defense-related genes, thereby boosting the plant’s resistance to pathogens [72]. When the treated plant is infected by pathogens, SA concentration typically increases 10- to 20-fold to trigger the plant defense response [73]. Salicylic acid glucosyltransferase (SAGT) is responsible for converting SA into its inactive metabolites, 2-O-β-D-glucose and SA glucose ester. Thus, SAGT functions to lower SA levels and maintain homeostasis in the plant host [73]. A model system using suspension-cultured cells of *A. thaliana* infected with the bacterial pathogen *Pst* DC3000 avrRpm1 was employed to screen for molecules that increase plant resistance without triggering plant resistance. Imprimatins A and B were identified from this model system; these compounds can elevate SA levels in A. thaliana plants after inoculation with either avirulent or virulent bacterial strains, subsequently enhancing plant resistance (Figure 4a–e) [74]. At concentrations effective for inducing plant disease resistance, imprimatin compounds can inhibit the activity of SAGT mutants in vitro. The results suggest that SAGT can competitively bind with imprimatins to SA [73].

Similar to SA, imprimatins C1 and C2 induce cell death in *A. thaliana* cells infected with avirulent *Pst*-*avrRpm1*, demonstrating concentration-dependent effects (Figure 5m,n). These compounds, collectively referred to as “imprimatin C” molecules, induce the accumulation of *PR-1* mRNA in both wild-type and *salicylic acid-induced deficient 2* (*sid2*) mutant *A. thaliana* seedlings, independent of pathogen infection. However, pretreatment with imprimatin C1 failed to induce SA accumulation in *A. thaliana* cells, suggesting that its action operates downstream of SA in the SA signaling pathway and does not activate SA biosynthesis through positive feedback cycles [34,75].

##### Diuretics

Certain diuretic compounds have been found to act as plant activators by inducing the SA signaling pathway. These compounds enhance the plant’s defense responses, making them more resistant to various diseases. Three potential diuretics—3-(butylamino)-4-phenoxy-5-sulfamoylbenzoic acid (bumetanide, BMT) (Figure 4f), 3-benzyl-1,1-dioxo-6-(trifluoromethyl)-3,4-dihydro-2H-1,2,4-benzothiadiine-7-sulfonamide (bendroflumethiazide, BFM) (Figure 4g), and 4-chloro-N-(2,6-dimethyl-1-piperidyl)-3-sulfamoylbenzamide (clopamide, CLP) (Figure 4h)—were identified from the model system of suspension-cultured A. thaliana cells described earlier. These compounds promoted cell death caused by *Pst* avrRpm1 in a concentration-dependent manner, although their effects were weaker than those of SA. All three compounds could enhance the resistance of A. thaliana seedlings against both wild-type *Pst avrRpm1* and *Pst*. Unlike imprimatins A and B, BMT and BFM mildly inhibited the activity of SAGT, while CLP did not inhibit SAGT. These upstream activators are crucial for the activation of the SA signaling pathway, which is essential for the plant’s immune response [76].

#### 2.1.2. Downstream Activators

Downstream activators include BTH (Figure 5a), VAA (Figure 5b), N-cyanomethyl-2-chloroisonicotinamide (NCI), 2,6-dichloroisonicotinic acid (INA) (Figure 5c), tiadinil (TDL) (Figure 5d), 4-methyl-1,2,3-thiadiazole-5-carboxylic acid (SV03) (Figure 5e), 3-chloro-1H-pyrazole-5-carboxylic acid (CMPA) (Figure 5f), benzoylsalicylic acid (BzSA) (Figure 5g), acetylsalicylic acid (ASA) (Figure 5h), 3,5-dichlorosalicylic acid (3,5-DCSA) (Figure 5i), 4-chlorosalicylic acid (4-CSA) (Figure 5j), 5-chlorosalicylic acid (5-CSA) (Figure 5k), dichloroanthranilic acid (DCA) (Figure 5l), imprimatins C1 (Figure 5m), and imprimatins C2 (Figure 5n), all of which play critical roles in activating the SA signaling pathway and enhancing plant immunity.

##### BTH

BTH, chemically known as S-methyl benz [1,2,3] thiadiazole-7-carbothioate, was first reported by Kunz et al. (Figure 5a) [77]. Also known as acibenzolar-S-methyl, BTH is a synthetic analog of SA that acts downstream of the SA signaling pathway. It mimics SA effects, inducing the expression of defense genes and enhancing disease resistance. Subsequent studies systematically explored various bioassay models for plant activators and found that the cucumber–*Colletotrichum lagenarium* model was superior in terms of resistance response and experimental stability compared to other plant–pathogen models, such as cucumber/*Pseudomonas lachrymans*, wheat/*Blumeria graminis* f.sp. *tritici*, and tomato/*P. infestans*. Field and greenhouse trials confirmed BTH’s high protective activity, leading to its development by Novartis/Syngenta under the trade name “Bion” [32,39]. BTH is metabolized by salicylic acid-binding protein 2 (SABP2) into acibenzolar, its active form, which induces SAR in various plants, including A. thaliana, tobacco, tomato, cucumber, wheat, sunflower, rice, maize, cotton, and soybean against multiple pathogens such as TMV, turnip crinkle virus, wheat powdery mildew, *Peronospora sparsa*, *Ps. syringae*, *F. oxysporum*, *Sclerotinia sclerotiorum*, and cucumber mosaic virus [39]. BTH also inhibited nematode reproduction, specifically *Meloidogyne incognita,* when applied by root dipping [78]. In tomatoes, BTH increased root cell wall density, reducing *F. oxysporum* entry [79]. In strawberries, BTH pretreatment increased phenolic compound concentrations in leaves infected by powdery mildew [80]. BTH induced increased abundance of *PR-1*, *PR-2*, and *PR-5* mRNAs in *A. thaliana* leaves [81]. It also upregulated SA pathway-related proteins in *Arctic bramble* (*Rubus arcticus*) and induced PAL and PR accumulation in rice [82,83]. BTH induced expression of *PR* genes in ethylene- and methyl jasmonate-insensitive mutant Arabidopsis plants, indicating that BTH operates independently of the JA and ET signaling pathways. Additionally, BTH increased *PR-1* gene expression in *NahG* transgenic tobacco, suggesting that its action occurs downstream of SA (Figure 6) [81,84]. In studies of BTH’s mode of action, acibenzolar was shown to induce expression of the *PR-1a* gene in SABP2-silenced tobacco, confirming SABP2 as a metabolic target [39]. In rice, BTH pretreatment upregulated *OsWRKY45* expression, a gene encoding transcription factor responsible for *PR* gene expression, indicating that acibenzolar, as the active form of BTH, could induce WRKY to upregulate *PR* gene expression [85].

##### NCI

NCI was first synthesized and identified as exhibiting protective activity against the rice/*M. oryzae* model [86,87,88]. Subsequent studies demonstrated that NCI had broad-spectrum protective efficacy against various pathogens, including Pst and Xanthomonas oryzae, in several plant species [89]. NCI is known to increase the abundance of mRNAs for the defense-related genes PR-1, PR-2, and PR-5 in *A. thaliana* leaves without altering the levels of SA [90]. Similarly, NCI pretreatment induced the expression of these PR genes in both wild-type and NahG transgenic tobacco leaves [90]. Importantly, pretreatment with NCI did not alter the concentrations of free or total SA in tobacco leaves. These findings indicate that NCI acts downstream of the SA signaling pathway [90]. NCI induced disease resistance in NahG transgenic plants but not in the npr1 mutant. It could also induce PR gene expression in the etr1-1, ein2-1, and jar1-1 mutants. Thus, NCI activates SAR independently of ET and jasmonic acid (JA) by targeting steps between SA and NPR1 [89,90]. Additionally, derivatives of NCI, such as N-methyl-2-chloroisonicotinamide and N-propargyl-2-chloroisonicotinamide, exhibited greater protective activity than NCI against rice blast [87].

##### INA

INA, a derivative of isonicotinic acid, exhibited marked resistance-inducing properties in various plant species (Figure 5c) [36,91]. Originally observed in cucumbers infected with *C. lagenarium*, subsequent studies have extended its protective effects to a wide array of plant–pathogen interactions. These include defense responses in tobacco against TMV, *A. thaliana* against Pst, sugar beet against *Cercospora beticola*, rose against *Sphaerotheca pannosa*, barley against *B. graminis* f. sp. *hordei*, and soybean against *S. sclerotiorum*, among others [34,36]. Field trials with INA demonstrated its efficacy in enhancing resistance against pathogens such as Uromyces appendiculatus in beans and Alternaria macrospora in cotton seedlings [92]. Furthermore, INA treatments have shown positive effects on seed germination and plant growth, as evidenced by increased shoot and root lengths in INA-treated tomato seedlings [93]. It also upregulated PR gene expression in rice, soybean, and tomato, enhancing the activity of defense enzymes such as chitinase and β-1,3-glucanase [34].

Additionally, INA treatments have been associated with the upregulation of antioxidant enzymes such as ascorbate peroxidase (APX) and catalase (CAT), contributing to enhanced defense responses in plants [93]. Furthermore, INA has been shown to cause the accumulation of intercellular washing fluid (IWF) in sugar beet leaves, leading to increased defense activity against pathogens such as *C. beticola* [94]. At a molecular level, INA primed SAR in tobacco plants by inducing the upregulation of PR gene expression in NahG transgenic plants, thereby inhibiting TMV replication without altering SA concentration. However, this priming effect was impaired in *A. thaliana* mutants that lacked functional NPR1- or NIM1-encoded proteins [95]. INA enhances the SA signaling pathway by interacting with NPR1, which in turn facilitates a coordinated immune response involving NPR1, HAC1, and TGAs. This cooperative mechanism helps strengthen the plant’s defense system [96].

##### TDL

TDL, or N-(3-chloro-4-methylphenyl)-4-methyl-1,2,3-thiadiazole-5-carboxamide, was introduced and registered as a commercial fungicide for combating rice blast disease in Japan in 2003 (Figure 5d) [97]. Upon application, TDL is converted into its active metabolite, SV03, which primes SAR in the plant (Figure 5e) [98]. Although TDL did not directly inhibit pathogen proliferation in vitro, bioassays demonstrated its efficacy in inducing resistance against rice blast and inhibiting the proliferation of viral, bacterial, and fungal pathogens [99,100,101]. At the molecular level, TDL induced the expression of genes involved in the SA signaling pathway in tobacco leaves without altering SA levels, suggesting that its site of action operates downstream of the SA pathway [102]. Similarly, SV03 increased the abundance of *PR-1a*, *PR-2*, and *PR-5* mRNAs in *NahG* transgenic tobacco leaves and was effective at inhibiting *Pst* proliferation and reducing lesion size in tobacco leaves infected with TMV [98]. These findings underscore the potential of TDL and its active metabolite SV03 as effective tools for enhancing plant resistance against various pathogens and provide insights into their mode of action within the SA signaling pathway [98,102].

##### CMPA

A pyrazole derivative, CMPA, reduced the disease symptoms of rice bacterial leaf blight in a dose-dependent manner but did not inhibit the proliferation of *X. oryzae*. CMPA upregulated PBZ1 mRNA in rice and tobacco, thereby activating SAR. Nevertheless, CMPA did not induce SA accumulation in tobacco (Figure 5f) [103,104,105].

##### SA and Its Derivatives

Endogenous SA serves as a key regulator of the SA signaling pathway, influencing SAR against various plant pathogens [106]. External application of SA, particularly during early disease stages or at the seedling stage, has been shown to confer protective effects against a wide range of plant diseases caused by viruses, bacteria, and fungi [107]. Such exogenous SA treatment induces the upregulation of PR genes in tobacco, enhancing resistance against pathogens such as TMV [108,109]. A derivative of SA, BzSA, also known as 2-(benzoyloxy) benzoic acid, was isolated from the seed coats of Givotia rottleriformis (Figure 5g) [110,111]. Spraying BzSA was shown to reduce lesion number and size in tobacco leaves infected with TMV [111].

Comparative analyses revealed that BzSA exhibited greater efficacy than SA or ASA (Figure 5h) in regulating the expression of marker genes in the SA signaling pathway and inhibiting TMV proliferation [111]. Furthermore, injecting tobacco plants with ASA or benzoic acid derivatives was shown to induce PR mRNA production and enhance resistance to TMV [111]. These findings underscore the potential of SA derivatives such as BzSA and ASA as effective agents for enhancing plant resistance against pathogens, offering valuable insights into their mechanisms of action and their applications in crop disease management strategies (Figure 6).

3, 5-DCSA (Figure 5i), 4-CSA (Figure 5j), and 5-CSA (Figure 5k) were shown to enhance disease resistance in tobacco against TMV infection, with reductions in lesion size of 70.0 ± 16.0%, 76.0 ± 11.0%, and 66.0 ± 9.0%, respectively. However, their potency in inducing disease resistance was slightly weaker than that of SA, which achieved a reduction in lesion size of 80.3 ± 7.2% [112,113]. These SA analogs also induced the accumulation of PR-1 in tobacco leaves, though with slightly lower resistance-inducing effects compared to SA [113]. The chlorinated derivatives of SA offered potential advantages over SA itself, such as increased stability and more prolonged resistance activity [114,115,116]. 3, 5-DCSA, 4-CSA, and 5-CSA bind to NPR1 with similar or slightly higher affinity than SA [117]. However, their effectiveness and modes of action may differ, and further research is needed to fully understand their potential as plant activators targeting the SA signaling pathway.

##### DCA

DCA, identified through screening for the expression of late upregulation in response to the *Hyaloperonospora parasitica* (*LURP*) gene cluster as a defense gene marker in *A. thaliana*, has been found to enhance disease resistance against virulent strains of the oomycete H. parasitica and the bacterium Pst DC3000 (Figure 5l). DCA induced the accumulation of PR1 and WRKY70 mRNA in plants but did not lead to SA accumulation [118]. Notably, the resistance-inducing effect of DCA is independent of the NPR1-mediated pathway, as demonstrated in npr1 mutants of *A. thaliana*. However, this effect is blocked in WRKY70 knockout mutants, indicating that DCA acts downstream of the SA signaling pathway and is distinct from compounds such as BTH, NCI, and INA [119].

##### Oxycom™

Oxycom™, a form of peracetic acid, has been observed to elicit a robust resistance response against numerous plant pathogens, particularly affecting leaf, berry, and root diseases [120,121,122,123]. For example, pretreatment of tobacco leaves with Oxycom™ reduced symptoms caused by Pst infection, with its protective efficacy surpassing that of SA. Furthermore, Oxycom™ demonstrated a superior ability compared to exogenous SA, significantly enhancing the expression of PR-1a, PR-1g, and PR-3a mRNAs in tobacco leaves [121]. Moreover, Oxycom™ was found to rapidly induce the phosphorylation of salicylic acid-induced protein kinase (SIPK) within one hour of treatment [121]. There was a positively correlated dynamic trend between the expression of PR-1a mRNA and SIPK phosphorylation, suggesting that certain defense genes associated with the SA signaling pathway were mediated by SIPK. These findings underscore the potential of Oxycom™ as an effective inducer of plant defense mechanisms, offering insights into its mode of action and its significance in plant disease management strategies [121].

#### 2.1.3. Activators with Unclear Action Sites

These activators represent a group of compounds that enhance plant defense mechanisms against pathogens, though the specific pathways involved are not fully understood. Some plant activators have unclear action sites within the SA signaling pathway.

##### Laminarin

Laminarin is a β-1,3-glucan derived from brown algae that has been shown to induce defense responses in various plants. While laminarin is known to activate the SA signaling pathway, its exact point of action remains unclear. Some studies suggest that it acts upstream of SA biosynthesis, while others propose that it acts downstream of or in parallel with the SA signaling pathway [123].

##### Chitosan

Chitosan, a deacetylated derivative of chitin, induces SA accumulation and upregulates the expression of defense-related genes in plants. However, the molecular mechanisms by which chitosan triggers the SA signaling pathway are not fully understood. Some researchers suggest it may interact with plant receptors to initiate signal cascades leading to SA biosynthesis, while others propose it acts through modulating ROS levels. A study investigated how chitosan oligosaccharides (COSs), a type of chitosan derivative, influence the proteome of rice plants infected with Southern Rice Black-Streaked Dwarf Virus (SRBSDV) using label-free quantitative proteomics. This research provided insights into the proteomic changes associated with chitosan-induced disease resistance. Additionally, hexanoic acid, a natural compound found in plants, has been reported to activate the SA signaling pathway and induce resistance against pathogens [124]. However, the specific targets of hexanoic acid within the SA signaling network remain unclear. Studies suggest it acts upstream of SA biosynthesis, while others propose it affects downstream components like NPR1. Azelaic acid, a dicarboxylic acid produced in plants during pathogen infection, induces SA accumulation and associated defense responses [125]. However, the precise mechanisms by which azelaic acid activates the SA signaling pathway are not well understood [126]. Some researchers suggest it modulates lipid-derived signaling molecules, while others propose it has direct effects on SA biosynthesis or downstream signaling components [127].

##### Riboflavin

Riboflavin (vitamin B2) has been reported to induce resistance against pathogens in various plants, potentially by activating the SA signaling pathway [128,129,130]. However, the specific targets and mechanisms by which riboflavin modulates the SA signaling cascade remain unclear. Some studies suggest that riboflavin may act by modulating cellular redox status or interacting with unknown receptors. These examples illustrate that, while many compounds have been identified as potential activators of the SA signaling pathway, their precise action sites and molecular mechanisms are often not well characterized. Further research is needed to elucidate the specific targets and modes of action of these activators within the complex SA signaling network.

##### 3, 4-Dichloro-2′-cyano-1, 2-thiazole-5-carboxanilide (Isotianil)

Isotianil, initially considered a derivative of PBZ, was discovered through the screening of isothiazole compounds by Bayer CropScience AG (Leverkusen, Germany) and was subsequently developed as a plant activator in collaboration with the Japanese company Sumitomo Chemical Co., Ltd. (Tokyo, Japan) (Figure 7a). Exhibiting marked efficacy, isotianil demonstrated protective activity against various rice diseases, including rice blast and bacterial leaf blight [36,131].

##### Polypeptide Product

Polypeptide product, also known as “Pen”, is a water-soluble substance isolated from *Penicillium chrysogenum* in the 1990s [132]. Foliar spraying or soil drenching with Pen induces plant disease resistance against various pathogens [132]. Pretreatment with Pen increases the resistance of fruits and vegetables to Pseudoperonospora viticola, Uncinula necator (syn. *Erysiphe necator*), *Venturia inaequalis*, *P. infestans*, *C. lagenarium* (syn. *Pseudocercospora lagenarium*), *Pseudocercospora cubensis*, and *Pseudomonas destructor*, and reduces disease severity [132]. When acidic or neutralized dry mycelium extracts were applied as drenches to the roots of melon plants, the treatment also enhanced resistance against *F. oxysporum* f. sp. melonis. These dry mycelium extracts increased POD activity in the plant host [133]. Nevertheless, the action site of Pen remains unclear.

##### β, γ-Methyleneadenosine 5′-triphosphate (AMP-PCP)

AMP-PCP, a non-hydrolyzable analog of ATP, enhances tobacco resistance against Pst and TMV [134]. Application of AMP-PCP increases the abundance of PR-1, PR-2, and PR-5 mRNAs in tobacco cv. Xanthi-nc leaves and decreases the concentration of extracellular ATP (eATP) [134]. However, increasing ATP concentration blocks the transcription of PR-1 mRNA, suggesting that a lower concentration (or absence) of eATP primes plant SAR. The action mechanism may involve two aspects: (1) ATP blocks SA synthesis; (2) ATP promotes SA degradation. Both actions reduce SA concentration, so AMP-PCP, as an antagonist of ATP, can serve as a potential plant activator to upregulate the SA signaling pathway [134].

##### 2, 2-Dichloro-3, 3-dimethylcyclopropane Carboxylic Acid (DDCC)

DDCC shows promise in preventing rice blast disease caused by Magnaporthe oryzae (Figure 7b) [135,136]. Studies have demonstrated that root drenching with DDCC enhances the activity of POD in rice leaves inoculated with *M. oryzae* [135]. Moreover, pretreatment with DDCC induces the production of melanin compounds at infection sites of *P. oryzae*, effectively inhibiting pathogen growth [137]. DDCC significantly induces the accumulation of phytoalexins, specifically momilactones A and B, at inoculation sites and surrounding tissue in rice leaves infected by *M. oryzae*. Interestingly, these momilactones are scarcely detected in healthy rice leaves [135]. Momilactones A and B exhibit direct inhibitory effects on fungal growth in vitro, with half-maximal effective dose (ED_50_) values of 4.8 and 0.9 ng/mL, respectively. Additionally, these compounds promote hyphal disruption in liquid mediums, suggesting their role as specialized direct defense agents against pathogens [136]. However, the action mechanism of DDCC remains elusive and requires further research.

##### Hyaluronic Acid (HA)

HA, a linear, unbranched polyanionic disaccharide produced by vertebrates and bacteria, consists of alternating glucuronic acid (GlcUA) and N-acetyl glucosamine (GlcNAc) units joined by β-1-3 and β-1-4 glycosidic bonds [138]. Recent studies have highlighted its potential as a plant activator, showcasing its ability to enhance protective activity and mitigate disease severity in various plant species [138]. For instance, cucumbers infected by *Colletotrichum orbiculare* or *P. syringae* pv. *lachrymans*, tomatoes infected by Pst or *X. axonopodis* pv. *vesicatoria*, and peppers infected by cucumber mosaic virus exhibited greater resistance in response to HA treatment [138]. Although HA does not directly inhibit these pathogens in vitro, its application can induce SAR in plants. Furthermore, investigations using a tobacco model with transgenic *PR-1a* promoter-*GUS* and *PDF1.2* promoter-GUS reporter genes revealed that HA strongly upregulated these genes [138]. This observation suggests that HA activates both the SA- and JA-mediated signaling pathways, distinguishing it from other known plant activators [138]. This dual activation of SA and JA signaling pathways by HA underscores its potential as a versatile tool for enhancing plant defense mechanisms against a wide range of pathogens [138].

##### Compounds Containing Indole and 4, 5-Dihydro-1H-pyrazoline Structures

Bai et al. [139] described the development of new antiviral compounds containing indole and 4,5-dihydro-1H-pyrazoline. These novel compounds were tested against potato virus Y and demonstrated significant antiviral activity. Notably, one compound, D40, exhibited curative and protective effects superior to those of the commercial antiviral agent Ningnanmycin. D40 was shown to enhance key defense enzyme activities and regulate the carbon fixation pathway in plants [139].

##### Compounds Containing Pyrazole Structures

Research by Ouyang et al. [140] focused on the design and synthesis of novel pyrazole derivatives containing oxime ester groups. These compounds were evaluated for their antiviral activities and demonstrated significant protective effects. The study also explored structure–activity relationships to understand how modifications impact the induction of host plant resistance as antiviral agents.

Yang et al. [141] demonstrated significant antiviral activity against pepper mild mottle virus (PMMoV) with cytosine derivatives containing a sulfonamide moiety, which showed superior efficacy compared to standard antiviral agents. The study also included a 3D-QSAR (quantitative structure–activity relationship) model to understand the structure–activity relationships. This indicated the potential of 1,4-pentadien-3-one derivatives containing the 1,3,4-oxadiazole moiety for further antiviral development, although bioassays showed that many of the compounds exhibited only moderate inhibitory activity against TMV in vivo [141]. Additionally, benzofuran derivatives containing disulfide moieties demonstrated significant antibacterial activity by interfering with protein synthesis. This research highlights the potential of these novel derivatives as effective antibacterial agents [141].

## 3. Classification Based on Chemical Structure

Classifying plant activators based on their chemical structures helps in understanding their potential modes of action and interactions with the SA signaling pathway. Plant activators can be categorized into various classes.

### 3.1. Salicylates

Salicylates are a primary class of plant activators that includes SA and its derivatives. SA, a plant hormone, directly modulates the SA signaling pathway by binding to specific receptors and triggering downstream defense responses [106]. This interaction activates a cascade of molecular events leading to the expression of defense-related genes. Research has shown that SA induces the expression of PRs, which play a significant role in pathogen resistance [106]. Derivatives of SA, such as BzSA and ASA, also contribute to defense activation. BzSA mimics SA’s effects and enhances SAR, while ASA, commonly known as aspirin, is converted to SA in plants. Both BzSA and ASA help stabilize the SA signal and extend its activity. Salicylates are integral to the plant’s innate immune system, and their ability to modulate the SA pathway makes them effective at increasing plant disease resistance. Studies on these compounds have improved our understanding of how SA-related pathways contribute to plant immunity and have practical applications in agriculture [111].

### 3.2. Benzothiadiazoles

Benzothiadiazoles are synthetic compounds that mimic SA and activate SAR. A prominent example in this class is BTH, which is widely used in agricultural practices to induce plant immunity. BTH acts as an SA analog by binding to SA receptors and activating the SAR pathway. It induces the expression of PR proteins and other defense-related genes, similar to the action of SA itself [82,83]. Research has demonstrated that BTH application can enhance disease resistance in a variety of crops, including tomatoes, cucumbers, and wheat [32,39,79]. The effectiveness of BTH and other benzothiadiazoles in activating SAR is attributed to their ability to stabilize the SA signaling pathway and enhance the plant’s systemic defense responses. Benzothiadiazoles represent a class of plant activators that offer practical solutions for managing plant diseases. Their ability to induce resistance in different plant species highlights their potential for widespread use in crop protection.

### 3.3. Nicotinamide Analogs

Nicotinamide analogs, such as INA and NCI, are structurally related to nicotinamide and play significant roles in defense activation by mimicking SA and influencing the SA signaling pathway. INA, an SA analog, induces systemic resistance in plants by activating the same defense mechanisms as SA. It enhances the expression of PR genes and increases resistance to pathogens [34,93,94,95]. NCI, another nicotinamide analog, also contributes to plant immunity by inducing similar defense responses. The chemical similarity of these analogs to nicotinamide, a component of coenzymes involved in metabolic processes, suggests that they might interact with other signaling pathways or metabolic processes in addition to SA signaling [89,90,91]. Research on these analogs has expanded our understanding of how structurally related compounds can influence plant immunity and provide additional tools for enhancing plant defense.

### 3.4. Amino Acid Derivatives

Amino acid derivatives, such as (D,L)-3-aminobutyric acid (BABA), have unique structures and influence plant defense responses through mechanisms distinct from those of SA and its analogs. BABA is known for its ability to induce systemic resistance by priming the plant’s defense readiness. It triggers a pre-defense response that prepares plants to better cope with subsequent pathogen attacks [69,70,71]. BABA’s mode of action involves the activation of defense pathways that are independent of SA but can synergistically enhance overall plant immunity. Studies have shown that BABA induces defense responses by modulating various signaling pathways, including those related to JA and ET [70,71]. This broad-spectrum activation of defense mechanisms makes BABA a valuable tool for improving plant resistance. Amino acid derivatives offer an alternative approach to enhancing plant immunity, providing additional options for managing plant diseases [70,71].

### 3.5. Synthetic Chemicals with Different Structures

Synthetic chemicals that impact plant defense mechanisms through different pathways include various diuretics and other small molecules. These compounds are often designed to target specific aspects of plant immunity or to enhance the effectiveness of existing defense responses. Diuretics, while primarily used for managing fluid balance, have been found to influence plant defense by affecting metabolic pathways related to SA signaling. Other synthetic chemicals, including various small molecules, can modulate plant immune responses by interacting with signaling pathways or affecting the production of defense-related compounds [76]. The study of synthetic chemicals provides insights into novel approaches for enhancing plant immunity and developing new crop protection strategies. Their diverse mechanisms of action highlight the potential for creating targeted solutions to manage plant diseases and improve agricultural productivity.

In summary, classifying plant activators based on their chemical structures facilitates a better understanding of their modes of action and interactions with the SA signaling pathway. Each class of compounds offers unique benefits and applications, contributing to the advancement of plant protection strategies.

## 4. How Plant Receptors Trigger an Immune Response

Plants have evolved a sophisticated innate immune system to detect and respond to potential biotic threats. A key component of this system involves cell-surface PRRs, which are crucial for recognizing and responding to PAMPs or microbe-associated molecular patterns (MAMPs) [1]. PRRs, including receptor-like kinases (RLKs) and receptor-like proteins (RLPs), are specialized proteins located on the plant cell surface that specifically bind to these MAMPs/PAMPs. This interaction triggers pattern-triggered immunity (PTI), a primary immune response aimed at defending against a broad range of pathogens. The recognition of PAMPs/MAMPs by PRRs activates a series of intracellular signaling pathways, including MAPK cascades, leading to the expression of defense-related genes and the initiation of various immune responses. These pathways form the basis of PTI, setting the stage for further immune responses if the initial defense is inadequate [142,143]. One subgroup of these receptors is the leucine-rich repeat receptor-like proteins (LRR-RLPs), which play a crucial role in plant immunity. However, the mechanisms underlying ligand recognition and activation of LRR-RLPs remain elusive. A recent study reported the crystal structure of the LRR-RLP RXEG1 in *Nicotiana benthamiana*, which recognizes the XEG1 xyloglucanase enzyme from the pathogen *P. sojae* [142,143]. The structure revealed that RXEG1 recognized XEG1 primarily through its amino-terminal and carboxy-terminal loop-out regions. RXEG1, a receptor-like protein in plants, binds to XEG1, a glycoside hydrolase secreted by *P. sojae*, through interactions involving these loop-out regions. This structural insight highlights the specific regions of RXEG1 that are crucial for pathogen recognition, aiding in the understanding of the molecular mechanisms behind plant immunity and pathogen evasion strategies. The loops bound to the active site groove of XEG1, inhibiting its enzymatic activity and suppressing *P. sojae* infection in *N. benthamiana*. The binding of XEG1 promoted the association of the leucine-rich repeat (LRR) domain of RXEG1 with the co-receptor BAK1, also an LRR-type protein, through RXEG1 and the conserved LRRs. This association triggered RXEG1-mediated immune responses (Figure 1). Structural comparisons of apo-RXEG1 (LRR), XEG1–RXEG1 (LRR), and XEG1–BAK1–RXEG1 (LRR) complexes showed that XEG1 binding induced conformational changes in the N-terminal region of RXEG1 and increased structural flexibility in the BAK1-associating regions of RXEG1 (LRR). These changes allowed RXEG1 to undergo a fold-switching mechanism, facilitating the recruitment of BAK1, which is characterized by its LRR domain. This domain is crucial for its interactions in plant signaling pathways, particularly in immune responses and developmental processes. It has been reported that there is a conserved mechanism of ligand-induced heterodimerization of an LRR-RLP with BAK1, suggesting a dual function of LRR-RLPs in plant immunity: direct ligand recognition and co-receptor recruitment for signaling. BIR2 (BAK1-interacting receptor-like kinase 2) plays a significant role in plant immune responses by interacting with BAK1 (a leucine-rich repeat kinase) and other proteins to regulate signaling pathways involved in defense against pathogens. Meanwhile, BIR2, a receptor-like kinase, recognized AVRY567 located on *P. sojae* cell walls and subsequently initiated PTI responses in *N. benthamiana*. The E3 ubiquitin ligases SNIPER2a and SNIPER2b targeted BIR2 for degradation via the 26S proteasome pathway, negatively regulating PTI. BAK1 associated with BIR2 upon recognition of AVRY567 and protected BIR2 from degradation mediated by SNIPER2a/b. Knockdown of BAK1 led to increased degradation of BIR2 by SNIPER2a/b, compromising AVRY567-triggered PTI responses. The kinase activity of BAK1 was required for its protective role against BIR2 degradation. BAK1 appeared to protect BIR2 by phosphorylating it, thereby preventing SNIPER2a/b from ubiquitinating and targeting it for degradation. This study revealed a novel function for the co-receptor BAK1 in stabilizing the receptor-like kinase BIR2 by shielding it from E3 ligase-mediated degradation. This stabilization enabled BIR2 to effectively initiate and sustain pattern-triggered immune responses upon recognition of the fungal AVRY567 protein in *N. benthamiana* [144,145,146,147,148].

### 4.1. Historical Development of Plant Activators

The historical development of plant activators has significantly advanced our understanding of plant immunity and led to practical applications in agriculture. This progress can be traced through early discoveries, the study of screening methods, and an understanding of target interactions (Figure 8).

#### 4.1.1. Development of Screening Methods

As our understanding of plant immunity advanced, so did the methods for identifying and evaluating plant activators. Early screening methods relied on observing plant phenotypes or symptoms and conducting biochemical assays to assess the effects of various compounds on plant disease resistance. However, these approaches often fell short in revealing the mechanisms of action or targets of plant activators.

In the 1990s and 2000s, high-throughput screening methods emerged, allowing researchers to test a large number of compounds quickly and efficiently. Techniques such as bioluminescent reporter gene assays and real-time quantitative PCR enabled quantitative assessment of gene expression related to plant defense, facilitating the identification of plant activators [149].

Advances in genomics and proteomics further revolutionized the screening process. Microarrays and next-generation sequencing technologies allowed for comprehensive analysis of gene expression profiles and identification of key regulatory pathways involved in plant immunity. These advances provided deeper insights into how plant activators influence immune responses at the molecular level [150].

The screening of plant activators involves several methodologies to identify compounds that effectively enhance plant resistance. One common approach uses plants, combined with in vivo observation of symptoms or disease severity. Plants are treated with potential activators and then exposed to pathogens; any reduction in disease symptoms indicates the efficacy of the activator (Table 1). Additionally, the detection of effector markers from the SA signaling pathway is employed. Techniques such as real-time quantitative PCR or Western blotting measure the expression levels of key marker genes like PRs, which indicate the activation of the SA signaling pathway (Table 2) [151]. Distinguishing between action targets using transgenic or mutant plants is also an important method. Genetically modified plants with specific mutations or transgenic expressions help reveal where and how activators interact with the plant’s defense mechanisms [152]. Molecular biology screening models further expand these methods by employing high-throughput techniques such as gene expression profiling, proteomics, and metabolomics to comprehensively analyze the effects of potential activators at the molecular level [153,154]. Together, these screening methods provide a robust framework for identifying and understanding the modes of action of effective plant activators.

Future research aims to develop more potent and selective SA signaling pathway activators with minimal side effects on plant growth and development (Table 3). Targeted approaches, such as structure-based drug design and genomic studies, may facilitate the discovery of novel activators with specific modes of action. Integrating activators with other defense-priming strategies, such as JA-mediated induced systemic resistance (ISR) and RNA interference (RNAi) methodologies, could provide more robust and durable resistance against pathogens (Figure 2). Exploring the potential of activators in modulating plant–microbe interactions and facilitating beneficial symbioses is an emerging area of interest. Field trials and the commercial development of activators as biostimulants or plant protection products are expected to increase, promoting sustainable agricultural practices.

#### 4.1.2. Studies of Target–Activator Interactions

The study of interactions between plant activators and their targets requires an understanding of the SA signaling pathway, which is central to plant immunity. This pathway is significant for its role in SAR [1,2,3]. When a plant activator targets the SA signaling pathway, it typically enhances the production and accumulation of SA, which, in turn, activates downstream defense genes such as PRs. These proteins bolster the plant’s immune response, making it more resistant to subsequent pathogen attacks. Interaction studies often focus on how activators influence key components of the SA signaling pathway, including NPR1 and EDS1 [2]. By using techniques such as gene expression analysis, protein–protein interaction assays, and studies with mutant plants, researchers can elucidate how specific activators modulate the SA signaling pathway, providing insights into their bioactivity and potential effectiveness as plant defense enhancers (Table 4) [155].

#### 4.1.3. Advancements in Understanding Target–Activator Interactions

Recent research has focused on elucidating the precise mechanisms by which plant activators interact with target molecules and pathways. Early studies identified broad patterns of activation, but more recent work has aimed to uncover detailed molecular interactions. One major advancement has been the identification of PRRs and their role in recognizing PAMPs. Research has shown that plant activators can modulate PRR activity, thereby influencing the activation of PTI [147,173]. This understanding has led to the development of activators that specifically target PRRs to enhance plant immune responses. Another significant development has been the characterization of ETI, which involves the recognition of specific pathogen effectors by plant resistance proteins. Studies have revealed how activators can influence ETI pathways and enhance resistance by interacting with resistance proteins or modifying effector functions. Advances in molecular modeling and structural biology have also contributed to our understanding of how plant activators interact with their targets. Techniques such as X-ray crystallography and nuclear magnetic resonance spectroscopy have provided detailed structures of protein–ligand complexes, shedding light on how activators bind to and influence specific proteins involved in plant immunity [174]. Overall, the development of plant activators reflects a progression from whole-plant observations in early studies to an understanding of molecular interactions in later research. This evolution has led to more targeted and effective strategies for enhancing plant resistance and managing plant diseases, with ongoing research continuing to expand the applications of plant activators in agriculture.

## 5. Current State of Knowledge of Plant Activators against the SA Signaling Pathway

### 5.1. Research Trends

The research landscape of plant activators has been marked by growing interest in understanding their roles in plant immunity and disease management. Researchers are exploring various types of plant activators, including synthetic chemicals and biologically derived agents, to identify those that offer the greatest efficacy in enhancing plant defense mechanisms. A notable trend is the increased focus on the molecular mechanisms underlying the action of plant activators, particularly their interactions with signaling pathways involving SA, JA, or ET. Additionally, there is a strong emphasis on developing smart delivery systems that enhance the effectiveness of plant activators while minimizing their environmental impact. The integration of advanced technologies, such as nanotechnology and genome editing, is also becoming more prevalent, aiming to improve the precision and application of plant activators in agriculture. Overall, global research is advancing toward more targeted and sustainable approaches to harnessing plant activators for improved crop health.

In recent years, researchers have synthesized a series of derivatives based on benzo-1,2,3-thiadiazole-7-carboxylic acid, with notable findings (Figure 9a). One such derivative, benzo-1,2,3-thiadiazole-7-carboxylic acid 2-(2-hydroxybenzoxyl) ethyl ester, exhibited the ability to substantially elevate levels of the ROS and hydrogen peroxide (H_2_O_2_) and enhance the activity of PAL in Taxus chinensis suspension cell cultures. Moreover, novel derivatives of benzo-1,2,3-thiadiazole carboxylic esters were synthesized, showing promising efficacy against a spectrum of bacteria and fungi in crops like cucumber, rice, and maize (Figure 9b). Subsequent research focused on synthesizing benzo-1,2,3-thiadiazole-7-carboxylate derivatives containing fluorine, revealing that derivatives with specific functional groups displayed substantial protective activity against pathogens like Erysiphe cichoracearum and *C. lagenarium* (Figure 9c) [175,176].

Additionally, novel thiadiazole derivatives incorporating thiazole or oxadiazole moieties have been synthesized, demonstrating inducible protective activity against pathogens, as reported by Gan et al. (Figure 9d,e) [177]. Notably, our research team developed an amino phosphonate compound containing benzo-1,2,3-thiadiazole, named dufulin, which exhibited superior protective activity across various crops under greenhouse and field conditions (Figure 9f). Dufulin showed inhibitory effects on the replication of TMV and SRBSDV, while also enhancing the expression of PR-1a in tobacco leaves and the activity of PAL, POD, and polyphenol oxidase (PPO) in rice leaves or rice suspension cells [178,179]. Furthermore, the novel pyrimidine-like compound 5-(cyclopropylmethyl)-6-methyl-2-(2-pyridyl) pyrimidin-4-ol (PPA) emerged as a promising plant activator, exhibiting inhibitory effects similar to BTH against disease in *A. thaliana* caused by Ps. syringae pv. maculicola (Figure 9g). PPA was found to cause the accumulation of mRNAs related to genes in the SA signaling pathway in *A. thaliana* leaves while also promoting H_2_O_2_ accumulation in intracellular sites inoculated with the pathogen, suggesting a novel mechanism of action [180]. Finally, 3-acetonyl-3-hydroxyoxindole (AHO), a derivative of isatin isolated from *Strobilanthes cusia*, demonstrated efficacy in reducing lesion number and size caused by TMV or Erysiphe cichoracearum (Figure 9h). AHO was found to induce the accumulation of PR-1a mRNA in *N. tabacum* cv. Xanthi-nc leaves and increased SA concentration and PAL activity. However, AHO pretreatment did not lead to the accumulation of SA and PR-1a mRNA in NahG transgenic tobacco, suggesting a complex interaction with the SA signaling pathway [181].

### 5.2. Challenges Facing Molecular Target Research

Recent research has made significant strides in identifying and understanding the molecular targets of plant activators. These advances are crucial for enhancing induced plant disease resistance and for the rational design of new activators.

#### 5.2.1. Discovery of New Targets for Plant Activators

Some studies have demonstrated that silencing certain susceptibility genes (*S* genes), such as *REM1.3*, using virus-induced gene silencing (VIGS) in plants can enhance resistance to pathogens like *P. infestans*. This technique has proven effective in crops such as potatoes and soybeans, revealing crucial targets for genetic modifications to improve disease resistance [182]. By detecting PAMPs, it has been found that PRRs and NLRs play a significant role in plant immune systems. Advances in understanding these receptors have led to the identification of new molecular targets that can be manipulated to strengthen plant defenses against a broad range of pathogens [1,25].

#### 5.2.2. Discovery of New Mechanisms for Plant Resistance Responses

Recent work highlights the role of long non-coding RNAs (lncRNAs) in regulating plant immune responses [183,184]. For instance, silencing specific lncRNAs has resulted in increased resistance to fungal pathogens causing tea leaf spot, providing new molecular targets for enhancing crop resistance [185,186].

#### 5.2.3. New Research Methods for Studying the Targets of Plant Activators

CRISPR/Cas9, a gene-editing technology, has been employed to edit microRNAs (miRNAs), which are short non-coding RNAs involved in the post-transcriptional regulation of gene expression. Editing specific miRNAs, such as miR482b and miR482c in tomatoes, has shown promising results in increasing resistance to *P. infestans* [187]. Currently, this method can identify and modify critical genetic components involved in plant immunity, paving the way for targeted genetic enhancements [188].

#### 5.2.4. Crosstalk between the SA and JA Signaling Pathways and Their Synergistic Role in Disease Resistance

The interplay between these two pathways has been partially elucidated, revealing how their modulation can influence plant responses to biotic stress. Targeting these pathways and key nodes could aid in designing activators that fine-tune plant immune responses, providing better protection against diseases [189].

## 6. Future Goals of Plant Activators

### 6.1. Action Mechanisms of Plant Activators

Over the years, many researchers have isolated or developed numerous plant activators targeting the SA signaling pathway. However, several unresolved issues require further investigation.

Firstly, while some plant activators’ action sites have been identified as being upstream or downstream of the SA signaling pathway, the precise mechanisms and targets of these activators often remain unclear. For example, while SABP2 converts BTH into acibenzolar, the mechanism by which acibenzolar activates the SA signaling pathway and its target is not fully understood [39].

Secondly, certain plant activators with different molecular structures can target the same protein to induce plant resistance. Therefore, it is speculated that the precise mechanisms of these activators may differ. For instance, both imprimatins A and B target SAGT, but their interaction sites within SAGT differ [74].

Lastly, some plant activators can trigger the SA signaling pathway while also activating other signaling pathways or directly inhibiting pathogenic microorganisms. Thus, these compounds have diverse functions in the plant host, and their precise mechanisms require further research. Examples include BABA and HA [70,71,138].

### 6.2. Metabolism Mechanism of Plant Activators

The active components and metabolic mechanisms of certain plant activators remain unclear. Some activators are converted into specific metabolites within the plant host, which then induce plant resistance. However, other activators maintain disease resistance activity without a clear understanding of their actions and effects on metabolism. Examples include PBZ and its metabolite BIT, as well as TDL and its metabolite SV03 [68,98,102].

### 6.3. Screening Models of Plant Activators

The use of models, such as the *PR-1a* or *PDF1.2* promoters linked to the GUS reporter gene, has proven effective for screening plant activators, as has the CaBP22 promoter linked to the GUS gene. These models offer advantages in terms of stability and are capable of achieving precise intracellular localization of the active site [149,190]. Constructing models with the promoters of key genes upstream of the GUS gene will help identify the action sites of plant activators in the future.

### 6.4. Discovery of Lead Compounds Based on Novel Action Targets

Currently, we lack accurate information about the action targets of many plant activators, except for SABP and SAGT. As a result, we are unable to design a rational structural template targeting these molecular sites. The next step should be to investigate the action targets of these plant activators and screen for novel lead compounds from natural sources.

## 7. Conclusions

Plant activators hold exciting prospects for advancing our understanding of plant defense mechanisms and could revolutionize crop protection strategies. By elucidating the action mechanisms, metabolism, and screening models of plant activators, and by discovering lead compounds based on novel targets, researchers will be able to harness the full potential of these activators to mitigate crop losses and ensure global food security in the face of evolving biotic stresses.

## Figures and Tables

**Figure 1 genes-15-01237-f001:**
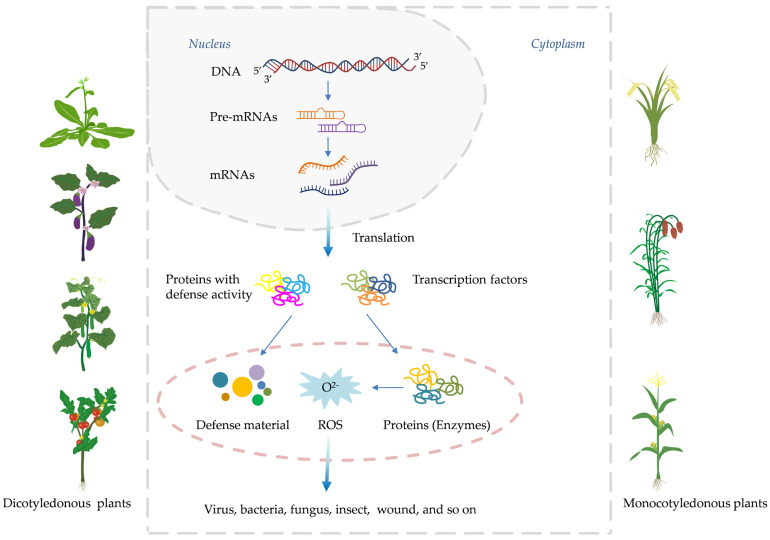
Defense mechanisms based on plant genetic regulation.

**Figure 2 genes-15-01237-f002:**
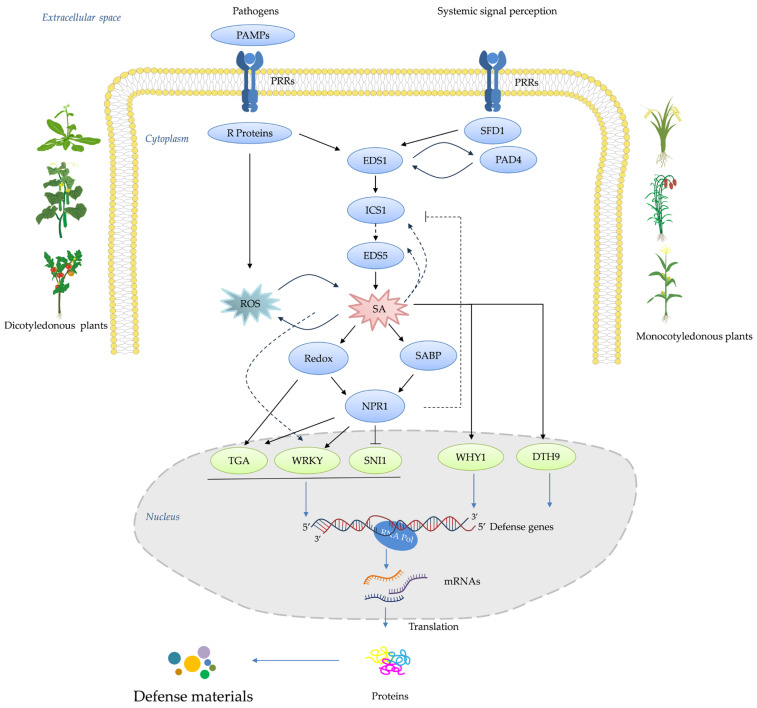
The salicylic acid signaling pathway and some related regulatory elements. Abbreviations: DTH9, detachment9; EDS, enhanced disease susceptibility; ICS1, isochorismate synthase; NPR1, nonexpressor of pathogenesis-related proteins 1; PAD4, phytoalexin biosynthesis 4; ROS, reactive oxygen species; SA, salicylic acid; SABP, salicylic acid-binding protein; SFD1, suppressor of fatty-acid-desaturase deficiency 1; SNI1, suppressor of Npr1-1 inducible 1; TGA, TGA-element-binding protein; and WHY1, WHIRLY1.

**Figure 3 genes-15-01237-f003:**
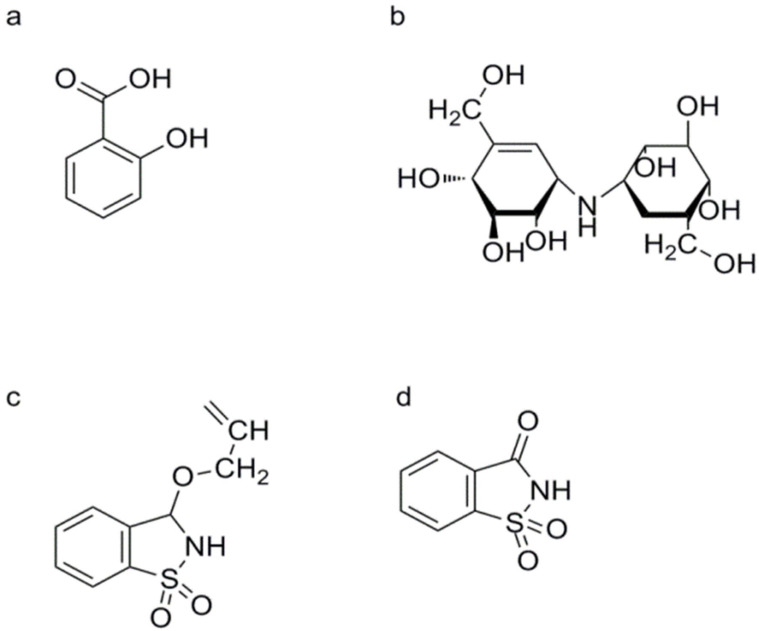
Salicylic acid (SA) and some molecules act upstream of the SA signaling pathway. (**a**) Salicylic acid, (**b**) validamycin A and validoxylamine A (VMA), (**c**) probenazole (PBZ), and (**d**) 1,2-benzisothiazol-3(2H)-1,1-dioxide (BIT).

**Figure 4 genes-15-01237-f004:**
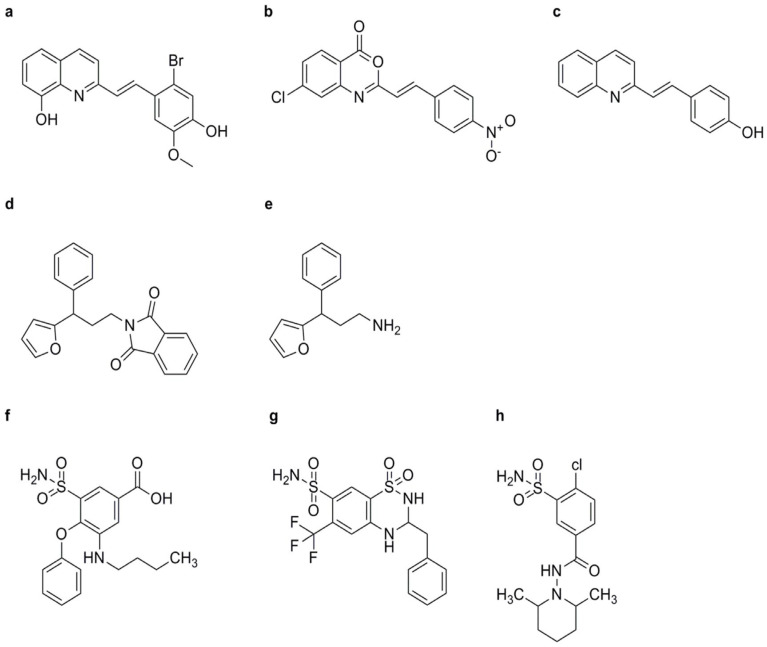
The molecular structure of imprimatins and diuretic compounds. (**a**) Imprimatin A1, (**b**) imprimatin A2, (**c**) imprimatin A3, (**d**) imprimatin B1, (**e**) imprimatin B2, (**f**) bumetanide, (**g**) bendroflumethiazide, and (**h**) clopamide.

**Figure 5 genes-15-01237-f005:**
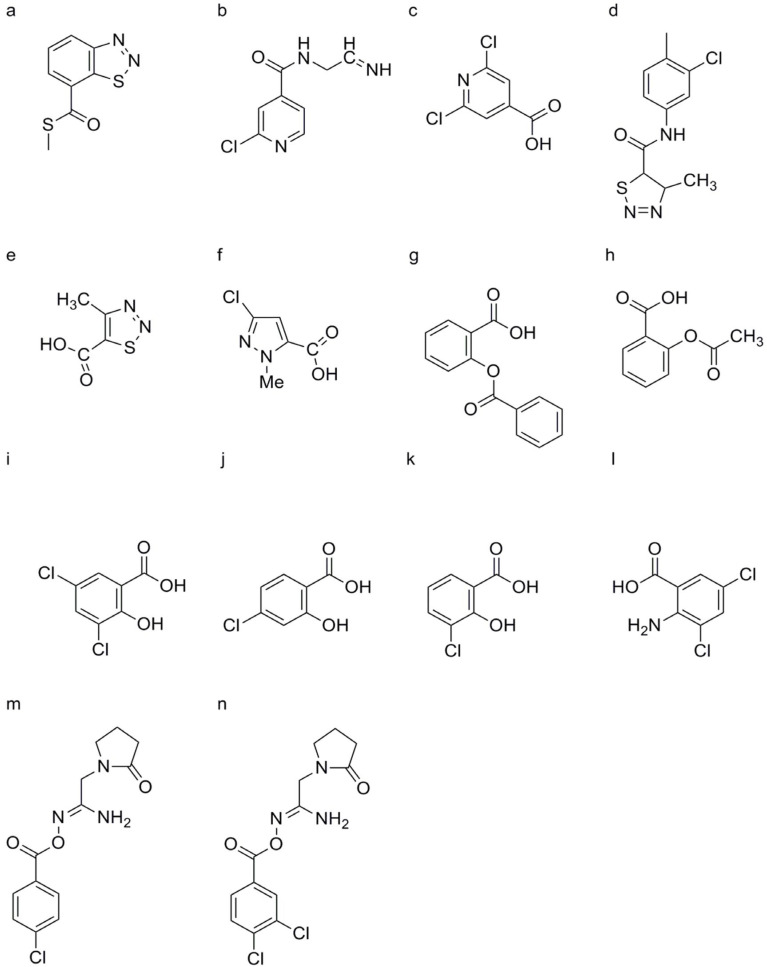
Some molecules act downstream of the SA signaling pathway. (**a**) Benzothiadiazole (benzo (1, 2, 3) thiadiazole-7-carbothioic acid S-methyl ester, BTH), (**b**) validoxylamine A (VAA), (**c**) 2,6-dichloroisonicotinic acid (INA), (**d**) tiadinil (TDL), (**e**) 4-methyl-1,2,3-thiadiazole-5-carboxylic acid (SV03), (**f**) 3-chloro-1H-pyrazole-5-carboxylic acid (CMPA), (**g**) benzoylsalicylic acid (BzSA), (**h**) acetylsalicylic acid (ASA), (**i**) 3, 5-dichlorosalicylic acid (3, 5-DCSA), (**j**) 4-chlorosalicylic acid (4-CSA), (**k**) 5-chlorosalicylic acid (5-CSA), (**l**) dichloroanthranilic acid (DCA), (**m**) imprimatins C1, and (**n**) imprimatins C2.

**Figure 6 genes-15-01237-f006:**
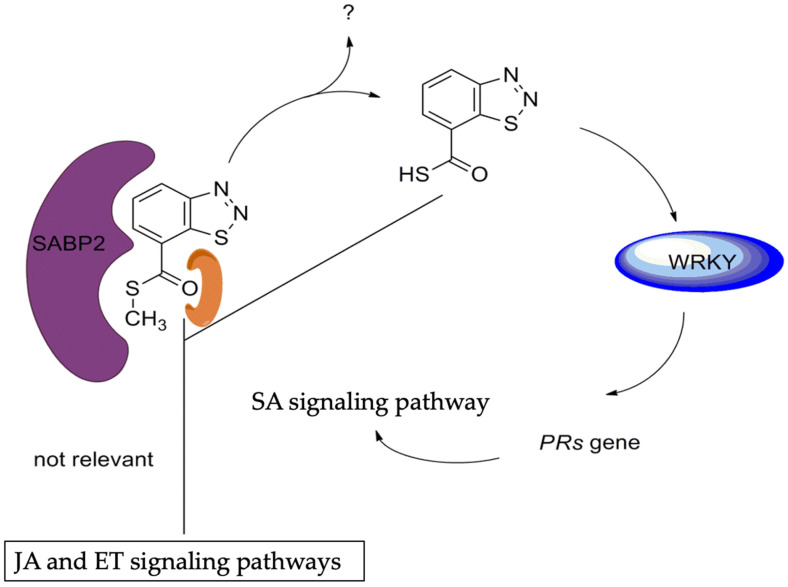
The action mechanism of benzothiadiazole. Abbreviations: BTH, benzothiadiazole; ET, ethylene; JA, jasmonic acid; PR, pathogenesis-related protein; and SABP2, salicylic acid-binding protein 2.

**Figure 7 genes-15-01237-f007:**
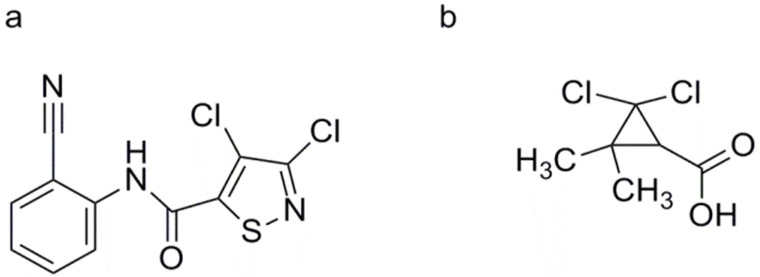
The molecule structures of (**a**) isotianil and (**b**) 2, 2-dichloro-3, 3-dimethylcyclopropane carboxylic acid (DDCC).

**Figure 8 genes-15-01237-f008:**
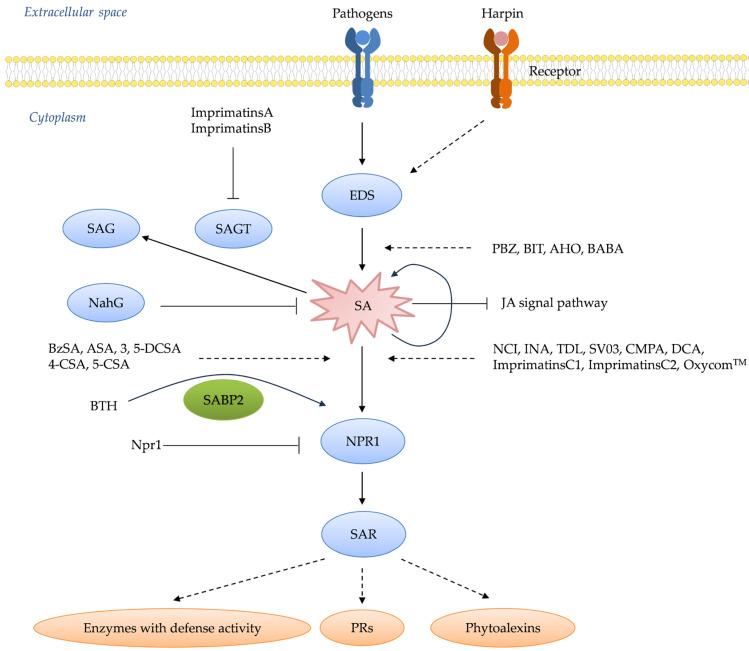
The action sites of some plant activators with respect to the salicylic acid signaling pathway.

**Figure 9 genes-15-01237-f009:**
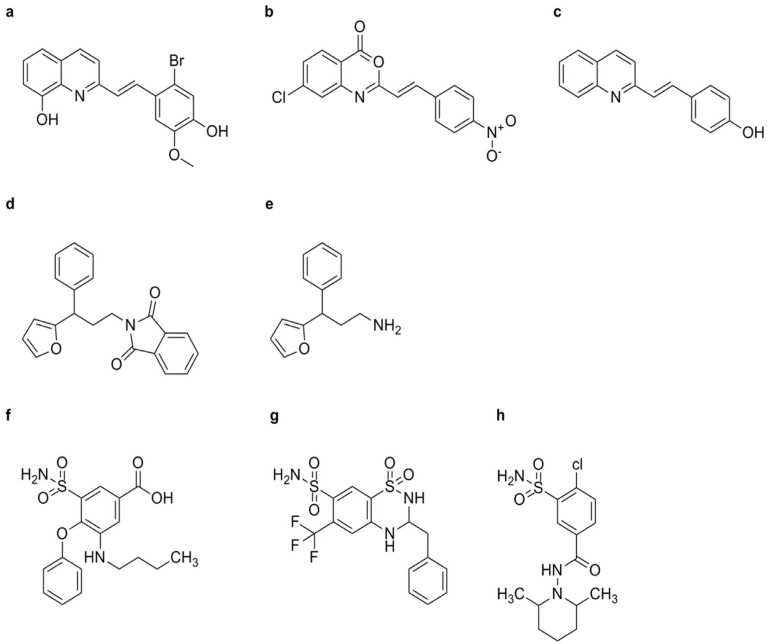
The plant activators targeting the SA signaling pathway developed in China. (**a**) Benzo-1, 2, 3-thiadiazole-7-carboxylic acid, (**b**) benzo-1, 2, 3-thiadiazole carboxylic esters, (**c**) benzo-1, 2, 3-thiadiazole-7-carboxylate derivatives, (**d**) thiadiazole derivatives incorporating thiazole, (**e**) thiadiazole derivatives incorporating oxadiazole moieties, (**f**) dufulin, (**g**) 5-(cyclopropylmethyl)-6-methyl-2-(2-pyridyl) pyrimidin-4-ol (PPA), and (**h**) 3-acetonyl-3-hydroxyoxindole (AHO).

**Table 1 genes-15-01237-t001:** The in vivo screening model for plant activators against multiple pathogens.

Plants	Plant Species	Pathogen Names	Evaluation Index	References
Cucumber	No mention	*C. lagenarium* (Paserini) Ell. & Halst	Infected leaf area	[32]
Tobacco	*N. tabacum* cv. Xanthi-nc	Tobacco mosaic virus	Lesion size	[155]
	*N. tabacum* cv. Xanthi-nc	*Oidium lycopersici*	Lesion size	[89]
	*N. tabacum* cv. Xanthi-nc	*P. syringae* pv. *tabaci*	CFU or disease severity	[155]
	*N. tabacum* cv. Xanthi-nc	*Cercospora nicotianae*	Infected leaf area	[155]
	*N. tabacum* cv. Xanthi-nc	*Peronospora tabacina*	Infected leaf area	[155]
	*N. tabacum* cv. Xanthi-nc	*Erwinia carotovora*	Length of infected stem	[155]
	*N. tabacum* cv. Xanthi-nc	*Phytophthora parasitica*	Disease severity	[155]
	*A. thaliana*	*Peronospora parasitica* pv. Emwa	Disease severity	[156]
Rice	*Oryza sativa* cv. Aichiasahi	*M. oryzae*	Lesion number	[89]
	*O. sativa* cv. Aichiasahi	*X. oryzae* pv. *oryzae*	Lesion length	[89]

**Table 2 genes-15-01237-t002:** The markers of defense response in the salicylic acid signaling pathway and the related methods involved.

Plant	Signal Pathways	Effector/Markers	Methods	References
Tobacco	Salicylic acid (SA)	Salicylic acid (free SA)	Organic solvent extract coupled with high-performance liquid chromatography	[15,33,154,157]
		SA (total SA)	Methanol extract coupled with high-performance liquid chromatography	[33]
		SA (salicylic acid-glucanase)	Methanol extract coupled with high-performance liquid chromatography	[15,157]
		*PR-1*	Northern blotting	[16,98,155,158]
		*PR-1*	Western blotting	[113,121,159]
		*PR-2*	Northern blotting	[16,98,155,158]
		*PR-3*	Northern blotting	[16]
		*PR-4*	Northern blotting	[16]
		*PR-5*	Northern blotting	[16,98,155,158]
		*PDF1.2*	Northern blotting	[160]
*Arabidopsis*	SA	*PR-1*	Northern blotting	[68,81]
	SA	*PR-2*	Northern blotting	[68,81]
	SA	*PR-5*	Northern blotting	[68,81]
Tomato	SA	*β*-1, 3-glucanase	Western blotting	[69]

**Table 3 genes-15-01237-t003:** The transgenic plant or mutant genes and their corresponding use in the activator detection methods.

Transgenics or Mutants	Plants	Action Sites	Signal Pathways	References
*NahG* transgenics	Tobacco	Salicylic acid (SA)	SA	[15,152,161]
*161SABP2*-silenced transgenic	Tobacco	SA level	SA	[39]
*nim* mutant	*A. thaliana*	Downstream of SA	SA	[156]
*npr1* mutant	*A. thaliana*	Downstream of SA	SA	[17]
*etr1* mutant	*A. thaliana*		Ethylene (ET)	[162]
*ein2* mutant	*A. thaliana*		ET	[163]
*jar1* mutant	*A. thaliana*		Jasmonic acid	[164,165]

**Table 4 genes-15-01237-t004:** Proteins that physically interact with salicylic acid.

Proteins Interacting with SA	Sources	Ligand Binding	References
Catalase	*N. tabacum*	^14^C-SA	[166]
Methyl esterase	*N. tabacum*	^3^H-SA	[167]
Carbonic anhydrase	*N. tabacum*	^3^H-SA	[168]
A protein with ankyrin repeat and BTB/POZ domain	*A. thaliana*	Genetics/T-DNA mutant	[117,169]
Cullin 3/CUL3 adapter protein	*A. thaliana*	Genetics/NPR1 paralog	[170]
Cullin 3/CUL3 adapter protein	*A. thaliana*	Genetics/NPR1 paralog	[170]
Ascorbate peroxidase	*N. tabacum*	^3^H-SA	[171]
E2 subunit of *α*-ketoglutarate dehydrogenase 2	*A. thaliana*	4-azido SA, SPR	[172]
Glutathione *S*-transferase 2	*A. thaliana*	4-azido SA, SPR	[172]
Glutathione *S*-transferase 8	*A. thaliana*	4-azido SA, SPR	[172]
Glutathione *S*-transferase 10	*A. thaliana*	4-azido SA, SPR	[172]
Glutathione *S*-transferase 11	*A. thaliana*	4-azido SA, SPR	[172]

## Data Availability

The data that support this study are available in the article.

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
