# Peer review of "The Past, Present, and Future of Plant Activators Targeting the Salicylic Acid Signaling Pathway"

_genes, 2024, doi:10.3390/genes15091237_

Round 1

Reviewer 1 Report

Comments and Suggestions for Authors

The main aims of this review are very interesting , both from a scientific and applicative point of view, because the discovery of innovative and more targeted molecules and approaches for plant defense is a very urgent issue.

However some essential points would need to be revised:

- please, update references

- Fig. 2 is too general, in particular concerning the recognition of elicitors/effectors, Fig.2 does not show anything about  PTI and ETI, and their specific receptors, it shows just citiplasmatic Receptor (ETI?), but this is a wrong picture of the plant immune system. While this subject is pivotal for a review dealing with plant activators.

- Still on PTI and ETI: PTI and PAMP/MAMP are mentioned after Fig.2, while to speak about PTI and ETI and their specific molecules is a priority to speak then about activators. For example, when speaking about "Plants possess cell-surface pattern recognition receptors that can detect danger signals from pathogens or the host itself, triggering an immune response [32]", this was the right point to introduce PTI, MAP/PAMP and their PRRs. Because Plant Recognition Receptors are the receptors for MAMPs/PAMPs, while here just LRRs are cited.

- still on plant immune responses, essential to specify that SAR can be induced both by PTI and ETI. Moreover, why a priori to exclude molecules from not pathogenic microbes triggering ISR? 

Author Response

Reviewer# 1

Comments and Suggestions for Authors

The main aims of this review are very interesting, both from a scientific and applicative point of view, because the discovery of innovative and more targeted molecules and approaches for plant defense is a very urgent issue. However, some essential points would need to be revised:

Response to Reviewer# 1

CommentsPlease, update the references

Response: Thank you, now we have updated references and modified main text.

Comments: Fig. 2 is too general, in particular concerning the recognition of elicitors/effectors, Fig.2 does not show anything about PTI and ETI, and their specific receptors, it shows just citiplasmatic Receptor (ETI?), but this is a wrong picture of the plant immune system. While this subject is pivotal for a review dealing with plant activators.

Response: According to your suggestion, we have updated Fig. 2.

Comments: Still on PTI and ETI: PTI and PAMP/MAMP are mentioned after Fig.2, while to speak about PTI and ETI and their specific molecules is a priority to speak then about activators. For example, when speaking about "Plants possess cell-surface pattern recognition receptors that can detect danger signals from pathogens or the host itself, triggering an immune response [32]", this was the right point to introduce PTI, MAP/PAMP and their PRRs. Because Plant Recognition Receptors are the receptors for MAMPs/PAMPs, while here just LRRs are cited.

Response: We have updated Fig. 2, and we have included suggested contents.

Comments:  still on plant immune responses, essential to specify that SAR can be induced both by PTI and ETI. Moreover, why a priori to exclude molecules from not pathogenic microbes triggering ISR? 

Response: Thank you for your suggestion. Systemic Acquired Resistance (SAR) can be induced by both Pattern-Triggered Immunity (PTI) and Effector-Triggered Immunity (ETI). PTI is generally triggered by the recognition of pathogen-associated molecular patterns (PAMPs), while ETI is often induced by the recognition of specific pathogen effectors. Both of these mechanisms can lead to the activation of SAR, which provides long-lasting protection against a broad range of pathogens. However, we have updated MS throughout and included some literature in revised MS.

Reviewer 2 Report

Comments and Suggestions for Authors

The manuscript “The past, present and future of plant activator targeting salicylic acid signal pathway” by Naz et al. (genes-3133426) is a review dealing with activators of the plant immune system (systemic acquired resistance or SAR) acting on the salicylic acid (SA) signal pathway. These activators are very interesting for their potential use as biological protectors of plants against pathogens and a review in this field appears useful. In their review, after a general introduction on SAR, the authors discuss the receptors triggering an immune response in plants and then the plant activators acting against the SA pathway, those acting downstream the pathway and those with unclear mechanism of action. The authors finally discuss activators recently studied in China and the future perspective in the field.

Although very rich in details and information, the manuscript requires a major revision because it appears rather unorganized, repetitive in many parts and confusing. The authors are suggested to deal with the different aspects of plant activators in a simpler and more organized way, dividing the activators in general categories and sections, according (for example) to their mechanism of action, chemical structure or other parameters. Redundancies in sentences and concepts should be avoided. The review would also be improved by a more concise and coordinated discussion of the topics.

A main reason for the lack of clarity in many sentences is apparently the fact that the manuscript requires a complete and careful English revision, because of many grammar and style errors, wrong terms, verbs and sentence structures, and so on.

Most figures also require revision because of low resolution, a lettering too small or misplaced, and legends missing key explanations of acronyms and data. In the text, the references to the figures are often unclear because sometimes they refer to the molecular structure and some other times to the mechanism of action.

The use of acronyms should be standardized: each molecule for which an acronym is used should be written in full the first time the molecule is mentioned in the text, with the acronym in brackets, then in the rest of the manuscript only the acronym must be used to indicate it.

Along the manuscript, the international rules of nomenclature should be applied. Authority and classification (order and family) should be provided for species and genera names when they are mentioned for the first time, then in the rest of the manuscript they should be indicated with abbreviations according to international rules of nomenclature. Binomial names should always be written in italics. Any common name of plants, animals and bacteria should be followed by the binomial name, with authority and classification, as previously mentioned.

Detailed comments

Title. Please replace “activator” with “activators”.

Introduction

Figure 1. The letters in the figure are too small and not clearly visible

Figure 2, legend. Please replace “SA” with “Salicylic acid (SA)”.

Lines 14, 194, 220, 235, 346, 374, 394, 415, 428, 430. Please replace “plant’s” with “plant”.

Lines 62-64. The sentence is unclear because the verb is missing: “SAR inducible” should be replaced with “The SAR is an inducible”.

Lines 67, 68, 72, 75, 204, 318, 329, 334, 441, 750. Delete the comma before “et al.” here and elsewhere in the manuscript.

Line 70. Perhaps “relevant” would be clearer than “seminal”.

Line 71. It is unclear what do the authors mean by “underpinning” when referring to SAR.

Lines 72-73. Replace “Malamy… “ with “Malamy et al.”

Line 77. It is unclear what do the authors mean by “in manifesting SAR”. Also, please replace “dissection” with “analysis”.

Line 85. Please replace “By harnessing our understanding of” with “Using the understanding obtained by”.

Line 93. Delete “the”.

Line 102. Please replace “burgeoning” with “promising”.

Figures 3 and 4. The figures show chemical structures at low resolution: the details are too small to be visible.

Figure 3, legend. Please replace “SA and some molecules act in SA upstream in SA signal pathway” with “SA and other molecules acting upstream in SA signal pathway”. Also, write in full the names of VMA, PBZ and BIT, followed by the abbreviation within brackets (see general comments on the review).

Lines 114, 115, 251, 260-262, 264-265, 267-269, 271, 293, 300, 308, 309, 351-352, 366-367, 376, 444-445, 450-453, 478, 480, 481, 483, 502-507, 517, 560, 565-566, 593-594, 595, 598, 680, 682-683, 686, 703-704, 722-724, 741, 747, 758-760, 763-765, 776.  Authority and classification should be provided for all species and genera names when they are mentioned in the manuscript for the first time, then in the rest of the manuscript they should be indicated with abbreviations according to international rules of nomenclature. For example, in line 114, “Nicotiana benthamiana” should be “Nicotiana benthamiana Domin (Solanales: Solanaceae), and from then on “N. benthaniana”. Furthermore, names of species and genera should always be written in italics (see general comments on the review).

Section 2

Line 118. “Phytophthora” should be in italics.

Lines 126-127. How is this sentence connected to Figure 1?

Lines 112-115, 122-131 and 136-141. These sentences are very confusing and disorganized. They should be rewritten in a simpler style, avoiding redundancies and explaining in a clearer way the triggering mechanism of plant activators. Each molecule indicated with an acronym must be transcribed in full the first time it is mentioned in the text, next to the acronym in brackets, then in the rest of the text only the acronym must be used to indicate the molecule (see general comments on the review).

Lines 142-143. See previous comments on acronyms. The line should read “the role of BAK1-interacting receptor-like kinase 2 (BIR2)”.

Line 158. The sentence should start like this “Here is a summary”.

Lines 169 and 174. Replace “the SA activators” with “SA activators”.

Line 176. What do the authors mean by “chemical genomics”? Furthermore, reference 52 does not appear to refer to plant activators.

Line 186. What sentence does “Figure 2” refer to?

Lines 187 and 191. Both paragraphs are indicated as “2.1”. Please check the paragraph numbering along the manuscript.

Line 189-190. It is unclear why lines 188-190 are entitled “The Past of Plant Activators” (Reference n. 58 is dated 2018).

In Table 1, each bacterial and fungal pathogen should be indicated by its binomial name, with authority and classification the first time it is used, then by abbreviations according to the international rules of nomenclature (see general comments on the review).

Line 227. Tables 1 and 2 should be clearly separated.

Line 229. The title of the paragraph should be “Studies on interactions between plant activators and their targets”

Table 2. This table should be completely reorganized. In column 1, each plant should be indicated by its binomial name, with authority and classification the first time it is used, then by abbreviations according to the international rules of nomenclature. Columns 2, 3 and 4 (“Signal pathways”, “Effector markers” and “Methods”, respectively) do not appear aligned with each other, so are difficult to understand. High performance liquid chromatography should be indicated the first time with its full name followed by its abbreviation, then always with abbreviation. In alternative, the abbreviations “HPLC” and others for Northern blotting and Western blotting (for example N and W in the column) could be indicated in the table legend.

Table 3. See comments on plant names in Table 2. What are the “detection methods” indicated in the table legend? Why are there no sites of action indicated in column 3, and what do “ET” and “JA” mean in column 4?

Table 4. Please replace “interacted” with “interacting” in the legend and title of column 1. In column 3, what is the meaning of the word “Genetics” related to ligands?

Line 253. Please delete “with”.

Line 254. Please replace “locating” with “located”.

Line 258. Please replace “characteristic of” with “characteristics of being”.

Line 259. Please delete “stability”.

Lines 279-280. Why is the name of MAPK in italics?

Lines 280 and 283. Please replace “plant the” with “the plant”.

Line 284. Please replace “mutants plant” with “mutant plants”.

Lines 287-290. These sentences are very confusing (because of language errors) and should be rewritten in a simpler and clearer way.

Line 301. What do the authors mean by “very effective against (Figure 2b)”?

Lines 291 and 301. Which one is the correct abbreviation for “Validoxylamine A”, “VMA” or “VAA”? Please define each abbreviation clearly and use them consistently. Do not use abbreviations in paragraph titles.

Lines 318-320. The sentence is unclear due to several errors and should be rewritten.

Line 323. What do the authors mean by “moral” in this sentence?

Lines 336-337. Provide binomial name, authority and classification (or at least orders and families, if names of genera and species are not available) for insects indicated only with the common name.

Lines 353-354 and 358. Why are the enzyme names in italics?

Lines 366, 369-370, 449-450, 537. Provide binomial name, authority and classification (or binomial names abbreviations) for species indicated only with the common name.

Lines 400, 404, 487, 503, 510, 520. “A. thaliana” should be in italics.

Line 403. Figure 3 only has letters from “a” to “d”: which figure does the citation refer to?

Figure 4. The letters are too small and the structures are too low resolution, difficult to visualize.

Line 417. By “Bumnide”, do the authors mean “Bumetanide”?

Line 427. What is the meaning of “nearly” in this sentence? 

Section 3

Line 432. Please replace “acted” with “acting”.

Line 438. Please replace “SA’s” with “SA”.

Line 440 and 530. Why is the name of the compound in italics?

Line 441. “schurter” should be “Schurter”.

Line 443. “colletotrichum legionarium” should be “Colletotrichum legionarium” in italics.

Lines 446-447. Why are company names in italics? Furthermore, can the names of companies and commercial products be directly mentioned in the manuscript?

Line 462. “Arctic bramble” is a common name: please provide binomial name (Rubus arcticus), authority and classification of this plant.

Figure 5. Each structure should be clearly identified in the legend with its chemical name and abbreviation. The letters are too small and often not clearly associated to a given structure. The figure should be completely reorganized.

Figure 6. The acronyms “SABP2” and “WRKY” should be explained in the figure legend.

Line 479, 481, 493, 496. Please replace “NCI’s” with “NCI”.

Line 524. Please replace “INA’s” with “INA”.

Line 531. See comment on lines 446-447.

Line 597. Please replace “DCA’s” with “DCA”.

Lines 601-610. Why are imprimatins C1 and C2 described in paragraph 3.1.9 and not in paragraph 2.2.5, which deals with generic imprimatins?

Figure 7. As previously mentioned for other figures, all acronyms should be explained in the figure legend.

Section 4

Why are the subheading in Section 4 italicized?

Lines 635-637. The sentence about laminarin appears unrelated to what is reported in Table 1: where is the connection?

Line 664. To which sentence does “Table 2” refer?

Lines 667-668. See comment on lines 446-447. Can the names of companies and commercial products be directly mentioned in the manuscript?

Figure 8, legend. Please replace “structure” with “structure”.

Line 689. Please replace “accurate” with “precise”.

Lines 691 and 716. To which sentence (or sentences) do “Figure 5” and Figure 6” refer?

Section 5

Lines 736 and 772. Please replace “strides” with “progresses”.

Figure 9. As previously mentioned for other figures, all structures should be identified by their complete names and abbreviations. Also, please delete “by” in the figure legend (line 770).

Line 796. Please replace “which showing the promising results” with “showing promising results”.

Figure 7. As previously mentioned for other figures, all acronyms should be explained in the figure legend.

Comments on the Quality of English Language

The manuscript requires a complete and careful revision of English style and grammar.

Author Response

Reviewer# 2

Comments and Suggestions for Authors

The manuscript “The past, present and future of plant activator targeting salicylic acid signal pathway” by Naz et al. (genes-3133426) is a review dealing with activators of the plant immune system (systemic acquired resistance or SAR) acting on the salicylic acid (SA) signal pathway. These activators are very interesting for their potential use as biological protectors of plants against pathogens and a review in this field appears useful. In their review, after a general introduction on SAR, the authors discuss the receptors triggering an immune response in plants and then the plant activators acting against the SA pathway, those acting downstream the pathway and those with unclear mechanism of action. The authors finally discuss activators recently studied in China and the future perspective in the field.

Although very rich in details and information, the manuscript requires a major revision because it appears rather unorganized, repetitive in many parts and confusing. The authors are suggested to deal with the different aspects of plant activators in a simpler and more organized way, dividing the activators in general categories and sections, according (for example) to their mechanism of action, chemical structure or other parameters. Redundancies in sentences and concepts should be avoided. The review would also be improved by a more concise and coordinated discussion of the topics.

A main reason for the lack of clarity in many sentences is apparently the fact that the manuscript requires a complete and careful English revision, because of many grammar and style errors, wrong terms, verbs and sentence structures, and so on.

Most figures also require revision because of low resolution, a lettering too small or misplaced, and legends missing key explanations of acronyms and data. In the text, the references to the figures are often unclear because sometimes they refer to the molecular structure and some other times to the mechanism of action.

The use of acronyms should be standardized: each molecule for which an acronym is used should be written in full the first time the molecule is mentioned in the text, with the acronym in brackets, then in the rest of the manuscript only the acronym must be used to indicate it.

Along the manuscript, the international rules of nomenclature should be applied. Authority and classification (order and family) should be provided for species and genera names when they are mentioned for the first time, then in the rest of the manuscript they should be indicated with abbreviations according to international rules of nomenclature. Binomial names should always be written in italics. Any common name of plants, animals and bacteria should be followed by the binomial name, with authority and classification, as previously mentioned.

Response to Reviewer# 2

We want to express our sincere gratitude for the time you took to share your remarkable comments and suggestions. Your insights will play a vital role in enhancing our MS, and we are truly thankful for your contribution. Please see our response thanks again.

Detailed comments

Comments:  Title. Please replace “activator” with “activators”.

Introduction

Response: We have replaced “activator” with “activators”.

Comments:  Figure 1. The letters in the figure are too small and not clearly visible

Response: We have replaced Figure 1. Letters size.

Comments:  Figure 2, legend. Please replace “SA” with “Salicylic acid (SA)”.

Response: We have replaced “SA” with Salicylic acid (SA)”.

Comments:  Lines 14, 194, 220, 235, 346, 374, 394, 415, 428, 430. Please replace “plant’s” with “plant”.

Response: We have replaced “plant’s” with “plant”.

Comments:  Lines 62-64. The sentence is unclear because the verb is missing: “SAR inducible” should be replaced with “The SAR is an inducible”.

Response: We have replaced “SAR inducible” with correct verb “The SAR is an inducible”.

Comments:  Lines 67, 68, 72, 75, 204, 318, 329, 334, 441, 750. Delete the comma before “et al.” here and elsewhere in the manuscript.

Response: We have deleted the comma before “et al.” in the thought manuscript.

Comments:  Line 70. Perhaps “relevant” would be clearer than “seminal”.

Response: We have replaced Perhaps “relevant” with “seminal”.

Comments:  Line 71. It is unclear what do the authors mean by “underpinning” when referring to SAR.

Response: We have replaced “underpinning” with correct word understanding.

Comments:  Lines 72-73. Replace “Malamy… “ with “Malamy et al.”

Response: We have corrected the format of citation ,and  replaced“ Malamy with  Malamy et al.

Comments:  Line 77. It is unclear what do the authors mean by “in manifesting SAR”. Also, please replace “dissection” with “analysis”.

Response: We have corrected, wrong words, as dissection with SAR.

Comments: Line 85. Please replace “By harnessing our understanding of” with “Using the understanding obtained by”.

Response: We have corrected, wrong words, as understanding obtained by.

Comments:  Line 93. Delete “the”.

Response: We have deleted “the”.

Comments:  Line 102. Please replace “burgeoning” with “promising”.

Response: We have replaced “burgeoning” with “promising”.

Comments:  Figures 3 and 4. The figures show chemical structures at low resolution: the details are too small to be visible.

Response: We have replaced Figures 3 and 4. The figures with high resolution and now details are visible to see.

Comments:  Figure 3, legend. Please replace “SA and some molecules act in SA upstream in SA signal pathway” with “SA and other molecules acting upstream in SA signal pathway”. Also, write in full the names of VMA, PBZ and BIT, followed by the abbreviation within brackets (see general comments on the review).

Response: We have written full name of molecules and abbreviation within brackets.

Comments:  Lines 114, 115, 251, 260-262, 264-265, 267-269, 271, 293, 300, 308, 309, 351-352, 366-367, 376, 444-445, 450-453, 478, 480, 481, 483, 502-507, 517, 560, 565-566, 593-594, 595, 598, 680, 682-683, 686, 703-704, 722-724, 741, 747, 758-760, 763-765, 776.  Authority and classification should be provided for all species and genera names when they are mentioned in the manuscript for the first time, then in the rest of the manuscript they should be indicated with abbreviations according to international rules of nomenclature. For example, in line 114, “Nicotiana benthamiana” should be “Nicotiana benthamiana Domin (Solanales: Solanaceae), and from then on “N. benthaniana”. Furthermore, names of species and genera should always be written in italics (see general comments on the review).

Response: We have tried our best to follow your suggestions.

Section 2

Comments:  Line 118. “Phytophthora” should be in italics.

Response: Now “Phytophthora” in italics.

Comments:  Lines 126-127. How is this sentence connected to Figure 1?

Response: Now legend is for Figure 1. Plant defense responses based on genetic variation and availability.

Comments:  Lines 112-115, 122-131 and 136-141. These sentences are very confusing and disorganized. They should be rewritten in a simpler style, avoiding redundancies and explaining in a clearer way the triggering mechanism of plant activators. Each molecule indicated with an acronym must be transcribed in full the first time it is mentioned in the text, next to the acronym in brackets, then in the rest of the text only the acronym must be used to indicate the molecule (see general comments on the review).

Response: Thank you; we have corrected English in all suggested lines.

Comments:  Lines 142-143. See previous comments on acronyms. The line should read “the role of BAK1-interacting receptor-like kinase 2 (BIR2)”.

Response: We have corrected sentence. (The study investigates the role of BAK1-interacting receptor-like kinase 2 (BIR2). BIR2, also known as BAK1-interacting receptor-like kinase 2, plays a significant role in plant immune responses by interacting with BAK1(a leucine-rich repeat kinase) and other proteins to regulate signaling pathways involved in defense against pathogens. Meanwhile, BIR2 is a receptor-like kinase that recognizes the fungal cell wall protein AVRY567 to initiate pattern-triggered immunity (PTI) responses in Nicotiana benthamiana).

Comments:  Line 158. The sentence should start like this “Here is a summary”.

Response: We have started from “Here is a summary”.

Comments: Lines 169 and 174. Replace “the SA activators” with “SA activators”.

Response: We have made the correction.

Comments: Line 176. What do the authors mean by “chemical genomics”? Furthermore, reference 52 does not appear to refer to plant activators.

Response: We have corrected the sentence.

Comments: Line 186. What sentence does “Figure 2” refer to?

Response: We have cited figures1-2 in the introduction.

Comments: Lines 187 and 191. Both paragraphs are indicated as “2.1”. Please check the paragraph numbering along the manuscript.

Response: We have corrected the number of sections.

Comments: Line 189-190. It is unclear why lines 188-190 are entitled “The Past of Plant Activators” (Reference n. 58 is dated 2018).

Response: We have updated sentence and heading, The Past of Plant Activators enhancing plant defense.

Comments: In Table 1, each bacterial and fungal pathogen should be indicated by its binomial name, with authority and classification the first time it is used, then by abbreviations according to the international rules of nomenclature (see general comments on the review).

Response: We have updated information accordingly.

Comments:  Line 227. Tables 1 and 2 should be clearly separated.

Response: Thank you, now Table 2 is separated from Table 1.

Comments: Line 229. The title of the paragraph should be “Studies on interactions between plant activators and their targets”.

Response: We have modified information.

Comments:  Table 2. This table should be completely reorganized. In column 1, each plant should be indicated by its binomial name, with authority and classification the first time it is used, then by abbreviations according to the international rules of nomenclature. Columns 2, 3 and 4 (“Signal pathways”, “Effector markers” and “Methods”, respectively) do not appear aligned with each other, so are difficult to understand. High performance liquid chromatography should be indicated the first time with its full name followed by its abbreviation, then always with abbreviation. In alternative, the abbreviations “HPLC” and others for Northern blotting and Western blotting (for example N and W in the column) could be indicated in the table legend.

Response: Thank you, we have modified suggested information.

Comments: Table 3. See comments on plant names in Table 2. What are the “detection methods” indicated in the table legend? Why are there no sites of action indicated in column 3, and what do “ET” and “JA” mean in column 4?

Response: Information in the table heading has been modified.

Comments:  Table 4. Please replace “interacted” with “interacting” in the legend and title of column 1. In column 3, what is the meaning of the word “Genetics” related to ligands?

Response: Thank you for your suggestion. In Table 4. We have replaced “interacted” with “interacting” in the legend and title of column 1. And refers to the study of genes and their role in influencing the binding and function of ligand-binding, we have modified column 3 heading.

Comments:  Line 253. Please delete “with”.

Response: We have deleted word “with”.

Comments:  Line 254. Please replace “locating” with “located”.

Response: We have replaced “locating” with “located”.

Comments:  Line 258. Please replace “characteristic of” with “characteristics of being”.

Response: Thank you, we have replaced “characteristic of” with “characteristics of being”.

Comments:  Line 259. Please delete “stability”.

Response: We have deleted word “stability”.

Comments:  Lines 279-280. Why is the name of MAPK in italics?

Response: We have corrected now MAPK is in non-italic format.

Comments:  Lines 280 and 283. Please replace “plant the” with “the plant”.

Response: We have replaced “plant the” with “the plant”.

Comments:  Line 284. Please replace “mutants plant” with “mutant plants”.

Response:  Thank you we have replaced “mutants plant” with “mutant plants”.

Comments:  Lines 287-290. These sentences are very confusing (because of language errors) and should be rewritten in a simpler and clearer way.

Response: We have corrected sentence.

Comments:  Line 301. What do the authors mean by “very effective against (Figure 2b)”?

Response: We have corrected the sentence (very effective against various bacterial infections).

Comments:  Lines 291 and 301. Which one is the correct abbreviation for “Validoxylamine A”, “VMA” or “VAA”? Please define each abbreviation clearly and use them consistently. Do not use abbreviations in paragraph titles.

Response: Very valuable comments, thank you, we have corrected .Validamycin A (VMA ) and Validoxylamine A (VAA).

Comments:  Lines 318-320. The sentence is unclear due to several errors and should be rewritten.

Response: Thank you, we have corrected sentence:  The above results indicated that VMA and VAA act upstream of SA in the SA signaling pathway [106]. Yang et al. [110] demonstrated significant antiviral activity against the pepper mild mottle virus (PMMoV), with compound A32 showing superior efficacy compared to standard antiviral agents.

Comments:  Line 323. What do the authors mean by “moral” in this sentence?

Response: We have corrected word normal. 

Comments:  Lines 336-337. Provide binomial name, authority and classification (or at least orders and families, if names of genera and species are not available) for insects indicated only with the common name.

Response: We have included suggested information.

Comments:  Lines 353-354 and 358. Why are the enzyme names in italics?

Response: Now enzyme names not in italics.

Comments:  Lines 366, 369-370, 449-450, 537. Provide binomial name, authority and classification (or binomial names abbreviations) for species indicated only with the common name.

Response: We have update information.

Comments:  Lines 400, 404, 487, 503, 510, 520. “A. thaliana” should be in italics.

Response:  “A. thaliana” was revised in italics.

Comments:  Line 403. Figure 3 only has letters from “a” to “d”: which figure does the citation refer to?

Response: We have cited figure in the revised main text as shown in “Figure 3. Shows salicylic acid (SA) and some molecules act in salicylic acid upstream in salicylic acid signal pathway (a) salicylic acid. (b) Validamycin A and Validoxylamine A (VMA). (c) Probenazole (PBZ). (d) 1, 2-Benzisothiazol-3(2H)-1, 1-dioxide (BIT).”

Comments:  Figure 4. The letters are too small and the structures are too low resolution, difficult to visualize.

Response: We have enhanced the Resolution and letters size of Figure 4.

Comments:  Line 417. By “Bumnide”, do the authors mean “Bumetanide”?

Response: We corrected word Bumetanide.

Comments:  Line 427. What is the meaning of “nearly” in this sentence? 

Response: We have deleted “nearly” in this sentence.

Section 3

Comments:  Line 432. Please replace “acted” with “acting”.

Response: We have replaced acted” with “acting”.

Comments:  Line 438. Please replace “SA’s” with “SA”.

Response: We have replaced “SA’s” with “SA”.

Comments:  Line 440 and 530. Why is the name of the compound in italics?

Response: Now, the name of the compound not in italics.

Comments:  Line 441. “schurter” should be “Schurter”.

Response:  We have replaced “schurter” with “Schurter”.

Comments:  Line 443. “colletotrichum legionarium” should be revised in italics.

Response: Now it is in italics “Colletotrichum legionarium”.

Comments:  Lines 446-447. Why are company names in italics? Furthermore, can the names of companies and commercial products be directly mentioned in the manuscript?

Response: The names of companies have been removed from MS.

Comments:  Line 462. “Arctic bramble” is a common name: please provide binomial name (Rubus arcticus), authority and classification of this plant.

Response: We have updated information, Arctic bramble (Rubus arcticus (L.) R. & S.) belongs to the Rosaceae family within the order Rosales.

Comments:  Figure 5. Each structure should be clearly identified in the legend with its chemical name and abbreviation. The letters are too small and often not clearly associated to a given structure. The figure should be completely reorganized.

Response: Figure 5.  Legend has been explained. Figure 5. Some molecules act in salicylic acid downstream in salicylic acid signal pathway. (a) benzo (1, 2, 3) thiadiazole-7-carbothioic acid S-methyl ester (BTH). (b) Validamycin A and Validoxylamine A (VMA). (c) 2, 6-Dichloroisonicotinic acid (INA). (d) Tiadinil (TDL). (e) 4-methyl-1, 2, 3-thiadiazole-5-carboxylic acid (SV03). (f) 3-chloro-1H-pyrazole-5-carboxylic acid (CMPA). (g) benzoylsalicylic acid (BzSA). (h) acetylsalicylic acid (ASA). (i) 3, 5-dichlorosalicylic acid (3, 5-DCSA). (j) 4-chlorosalicylic acid (4-CSA). (k) 5-chlorosalicylic acid (5-CSA). (l) Dichloroanthranilic acid (DCA). (m) Imprimatins C1. (n) Imprimatins C2.

Comments:  Figure 6. The acronyms “SABP2” and “WRKY” should be explained in the figure legend.

Response: Figure 6. Legend has been explained. Figure 6.  The action mechanism of BTH. its  include: SABP2 (Salicylic Acid-Binding Protein 2), a key protein involved in the salicylic acid signaling pathway also  involved in plant defense responses and WRKY (WRKY Transcription Factor), Arrows indicating interactions or transformations between these elements. A family of transcription factors that play a critical role in regulating plant immune responses.

Comments:  Line 479, 481, 493, 496. Please replace “NCI’s” with “NCI”.

Response: We have replaced “NCI’s” with “NCI”.

Comments:  Line 524. Please replace “INA’s” with “INA”.

Response: We have replaced “INA’s” with “INA”.

Comments:  Line 531. See comment on lines 446-447.

Response: We have removed company name. Due to whole manuscript revision and modification the line numbers are might not same.

Comments:  Line 597. Please replace “DCA’s” with “DCA”.

Response: We have replaced “DCA’s” with “DCA”.

Comments:  Lines 601-610. Why are imprimatins C1 and C2 described in paragraph 3.1.9 and not in paragraph 2.2.5, which deals with generic imprimatins?

Response: Now paragraph 3.1.9 is a subsection that is 2.2.5.1.

Comments:  Figure 7. As previously mentioned for other figures, all acronyms should be explained in the figure legend.

Response: Figure 7. Legend has been explained. The action sites of some plant activators against salicylic acid signal pathway. SA: Salicylic acid, a key planthormone involved in defense responses. EDS: Enhanced disease susceptibility. SAG: Salicylic acid glucoside. SAGT: Salicylic acid glucosyltransferase. NahG: Salicylate hydroxylase. SABP2: Salicylic acid-binding protein 2. NPR1: Non expressor of pathogenesis-related genes 1. SAR: Systemic acquired resistance. PRs: Pathogenesis-related proteins. JA: Jasmonic acid. The image depicts a complex signaling pathway in plant defense, showing how plants respond to pathogens and other stress factors. It illustrates the interactions between various components in both the extracellular space and cytoplasm, culminating in the activation of defense mechanisms.

Section 4

Comments:  Why are the subheading in Section 4 italicized?

Response: We have corrected. Now these are non-italic.

Comments:  Lines 635-637. The sentence about laminarin appears unrelated to what is reported in Table 1: where is the connection?

Response: Thank you we have cited (Table 1) in appropriate place in the text.

Comments:  Line 664. To which sentence does “Table 2” refer?

Response: We have cited (Tables 2) in the appropriate place in the MS text.

Comments:  Lines 667-668. See comment on lines 446-447. Can the names of companies and commercial products be directly mentioned in the manuscript?

Response: We have deleted company names form MS.

Comments:  Figure 8, legend. Please replace “structure” with “structure”.

Response: we have replaced “structure” with “structure”. Figure 8. The molecule structure shows (a) Isotianil and (b) 2, 2-dichloro-3,3-dimethylcyclopropane carboxylic acid (DDCC).

Comments:  Line 689. Please replace “accurate” with “precise”.

Response:  We have replaced “accurate” with “precise”.

Comments:  Lines 691 and 716. To which sentence (or sentences) do “Figure 5” and Figure 6”

refer?

Response: “Figure 5” and Figure 6 we cited in main text. Figure 5. Some molecules act in the downstream in SA signaling pathway. (a) Benzothiadiazole (benzo (1, 2, 3) thiadiazole-7-carbothioic acid S-methyl ester, BTH). (b) Validamycin A (VMA) and Validoxylamine A (VAA). (c) 2,6-Dichloroisonicotinic acid (INA). (d) Tiadinil (TDL). (e) 4-methyl-1, 2, 3-thiadiazole-5-carboxylic acid (SV03). (f) 3-chloro-1H-pyrazole-5-carboxylic acid (CMPA). (g) benzoylsalicylic acid (BzSA). (h) acetylsalicylic acid (ASA). (i) 3, 5-dichlorosalicylic acid (3, 5-DCSA). (j) 4-chlorosalicylic acid (4-CSA). (k)5-chlorosalicylic acid (5-CSA). (l) Dichloroanthranilic acid (DCA). (m) Imprimatins C1. (n) Imprimatins C2. Figure 6. The action mechanism of Benzothiadiazole. Abbreviations: BTH, Benzothiadiazole; ET, ethylene; JA, jasmonic acid; PR, pathogenesis related protein; SABP2, Salicylic acid-binding protein 2.

Section 5

Comments:  Lines 736 and 772. Please replace “strides” with “progresses”.

Response: We have replaced strides” with “progresses”.

Comments:  Figure 9. As previously mentioned for other figures, all structures should be identified by their complete names and abbreviations. Also, please delete “by” in the figure legend (line 770).

Response: Figure 9 is ‘The plant activators targetting the salicylic acid signaling pathway developed in China. (a) Benzo-1, 2, 3-thiadiazole-7-carboxylic acid. (b) benzo-1, 2, 3-thiadiazole carboxylic esters.(c) Benzo-1, 2, 3-thiadiazole-7-carboxylate derivatives. (d) Thiadiazole derivatives incorporating thiazole. (e) Thiadiazole derivatives incorporating oxadiazole moieties. (f) Dufulin. (g) 5-(cyclopropylmethyl)-6-methyl-2-(2-pyridyl) pyrimidin-4-ol (PPA). (h) 3-acetonyl-3-hydroxyoxindole (AHO).

Comments:  Line 796. Please replace “which shows the promising results” with “showing promising results”.

Response: We have replaced “which shows the promising results” with “showing promising results”.

Comments:  Figure 7. As previously mentioned for other figures, all acronyms should be explained in the figure legend.

Response: All acronyms have explained in the figures legend.

Comments:  Comments on the Quality of English Language. The manuscript requires a complete and careful revision of English style and grammar.

Response: We have revised English style and grammar through MS. At present, the description of language in revised MS has been improved.